# Measurements of atmospheric ethene by solar absorption FTIR spectrometry

Geoffrey C. Toon, Jean-Francois L. Blavier, Keeyoon Sung
Jet Propulsion Laboratory, California Institute of Technology, CA 91109, USA

*Correspondence to:* Geoffrey.C.Toon@jpl.nasa.gov

**Abstract.** Atmospheric ethene ($C_2H_4$; ethylene) amounts have been retrieved from high-resolution solar absorption spectra measured by the Jet Propulsion Laboratory (JPL) MkIV interferometer. Data recorded from 1985 to 2016 from a dozen ground-based sites have been analyzed, mostly between 30N and 67N. At clean-air sites such as Alaska, Sweden, New
Mexico, or the mountains of California, the ethene columns were always less than $1x10^{15}$ molec.cm$^{-2}$ and therefore undetectable. In urban sites such as JPL, California, ethene was measurable with column amounts of $20x10^{15}$ molec.cm$^{-2}$ observed in the 1990s. Despite the increasing population and traffic in Southern California, a factor 3 decrease in ethene column density is observed over JPL in the past 25 years, accompanied by a decrease in CO. This is
likely due to Southern California's increasingly stringent vehicle exhaust regulations and tighter enforcement over this period.

## 1 Introduction

Atmospheric ethene arises from microbial activity in soil and water, biological formation in plants, and by incomplete combustion from sources such as biomass burning, power plants,
and combustion engines. Ethene is primarily destroyed by reaction with OH (Olivella and Sole, 2004), which is rapid, giving ethene a tropospheric lifetime of only 1 to 3 days. Despite covering only 29% of the Earth's area, the land produces 89% of the ethene (Sawada and Totsuka, 1986). This is mainly natural, but in urban environments or near fires, ethene from incomplete combustion can dominate. Sawada and Totsuka (1986) used measurements of ethene emissions
per unit biomass to derive a global source of 26.2 Tg yr$^{-1}$ from natural emissions and 9.2 Tg yr$^{-1}$ from anthropogenic emissions, giving a total of 35.4 Tg yr$^{-1}$, which ranges from 18-45 Tg yr$^{-1}$. Goldstein et al. (1996) measured ethene emissions from Harvard Forest, Massachusetts, and found that they were linearly correlated with levels of photosynthetically active radiation (PAR), indicating a photosynthetic source. Based on this, they estimated that at Harvard Forest biogenic
emissions of ethene correspond to approximately 50% of anthropogenic sources. Using these fluxes, and the ecosystem areas tabulated by Sawada and Totsuka (1986), a global biogenic

source for ethene of 21 Tg yr$^{-1}$ was calculated. This value is similar to the estimates of Hough (1991). The ethene fluxes listed by Poisson et al. (2000), however, are only 11.8 Tg yr$^{-1}$, while those of Muller and Brasseur (1995) are only 5 Tg yr$^{-1}$. Abeles et al. (1992) estimate a Terrestrial

biogenic source of 16.6 Tg yr$^{-1}$ and an anthropogenic source of 9.2 Tg yr$^{-1}$. Combustion of fossil fuels amounts to only 21% of these anthropogenic emissions globally, but in urban areas this can be the major source.

There have been previous measurements of ethene by in situ techniques and also by remote sensing. These will be discussed later in the context of comparisons with results from the

JPL MkIV interferometer, an infrared Fourier transform spectrometer that uses the sun as a source. We report here long-term remote sensing measurements of $C_2H_4$ in the lower troposphere, where the vast majority of $C_2H_4$ resides, by ground-based MkIV observations. We also present MkIV balloon measurements of $C_2H_4$ in the upper troposphere.

## 2  Methods

### 2.1  MkIV Instrument

The MkIV Fourier Transform Spectrometer (FTS) is a double-passed FTIR spectrometer designed and built at the Jet Propulsion Laboratory (JPL) in 1984 for atmospheric observations (Toon, 1991). It covers the entire 650-5650 cm$^{-1}$ region simultaneously with two detectors: a HgCdTe photoconductor covering 650-1800 cm$^{-1}$ and an InSb photodiode covering 1800-5650

cm$^{-1}$. The MkIV instrument has flown 24 balloon flights since 1989. It has also flown on over 40 flights of the NASA DC-8 aircraft as part of various campaigns during 1987 to 1992 studying high-latitude ozone loss. MkIV has also made 1132 days of ground-based observations since 1985 from a dozen different sites, from Antarctica to the Arctic, from sea-level to 3.8 km altitude. Details of the ground-based measurements and sites can be found at:

http://mark4sun.jpl.nasa.gov/ground.html. MkIV observations have been extensively compared with satellite remote sounders (e.g. Velazco et al., 2009) and with in situ data (e.g., Toon et al., 1999a,b).

### 2.2  Spectral Analysis

The spectral fitting was performed with the GFIT (Gas Fitting) code, a non-linear least-

squares algorithm developed at JPL that scales the atmospheric gas volume mixing ratio (vmr) profiles to fit calculated spectra to those measured. For balloon observations, the atmosphere was

discretized into 100 layers of 1 km thickness. For ground-based observations, 70 layers of 1 km thickness were used. Absorption coefficients were computed line-by-line assuming a Voigt lineshape and using the ATM linelist (Toon, 2014a) for the telluric lines. This is a "greatest hits" compilation, founded on HITRAN, but not always the latest version for every band of every gas. For example, in cases (gases/bands) where the HITRAN 2012 linelist (Rothman et al., 2012) gave poorer fits than HITRAN 2008, the earlier version was retained. The $C_2H_4$ linelist covering the 950 $cm^{-1}$ region containing the $v_7$ and $v_8$ bands, is described by Rothman et al. (2003). The solar linelist (Toon, 2014b) used in the analysis of the ground-based MkIV spectra was obtained from balloon flights of the MkIV instrument.

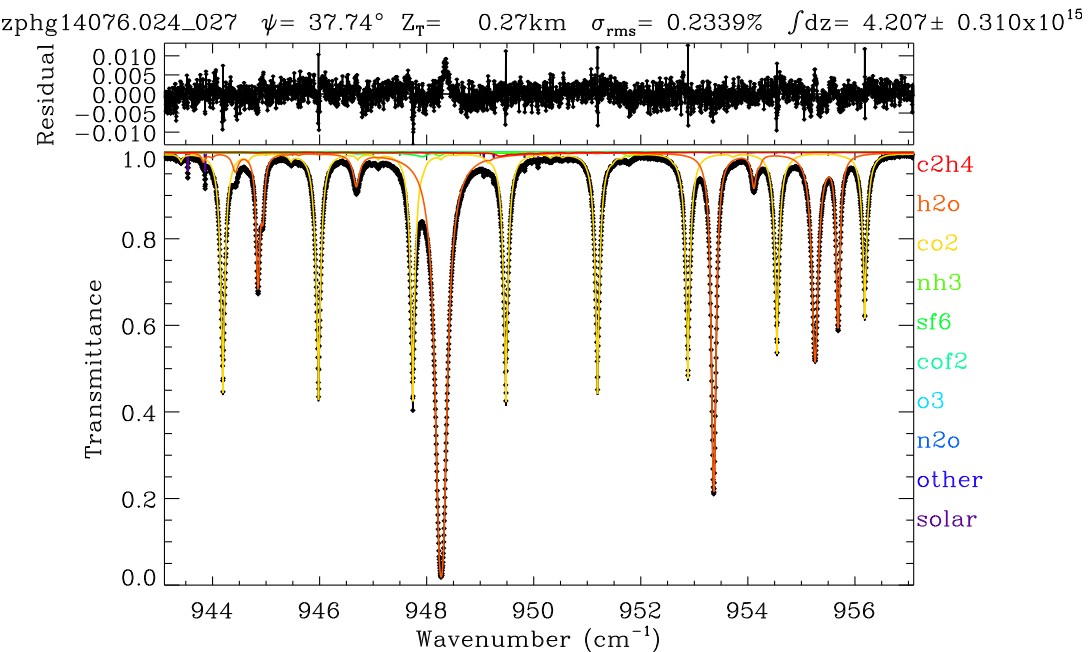

***Figure 1.*** *Example of a fit to a ground-based MkIV spectrum measured from JPL, California, on March 17, 2014 at a solar zenith angle of $\psi$ = 37.7° from a pressure altitude of $Z_T$ = 0.27 km. In the lower panel black diamond symbols represent the measured spectrum, the black line represents the fitted calculation, and the colored lines represent the contributions of the various absorbing gases; mainly $CO_2$ (amber) and $H_2O$ (orange). Also fitted are the 0% and 100% signal levels, separate telluric and solar frequency shifts, together with 5 more weakly absorbing gases ($NH_3$, $SF_6$, $COF_2$, $O_3$ and $N_2O$). The retrieved $C_2H_4$ column amount on this day, $4.2x10^{15}$ molec.$cm^{-2}$, would represent 2 ppb confined to the lowest 100 mbar (1.5 km) of the atmosphere. The $C_2H_4$ absorption contribution (red) peaks at 949.35 $cm^{-1}$ with an amplitude of less than 1% and therefore difficult to discern on this plot. The upper panel shows fitting residuals (measured - calculated), peaking at 1.3% with an rms deviation of 0.234%, which are mainly correlated with $H_2O$ and $CO_2$. The vertical column of $C_2H_4$ derived from this fit was $4.2\pm0.4x10^{15}$ molec.$cm^{-2}$.*

Sen et al. (1996) provide a more detailed description of the use of the GFIT code for retrieval of vmr profiles from MkIV balloon spectra. GFIT was previously used for the Version 3 analysis (Irion et al., 2003) of spectra measured by the Atmospheric Trace Molecule Occultation Spectrometer (ATMOS), and is currently used for analysis of TCCON spectra (Wunch et al., 2011) and MkIV spectra (Toon, 2016).

We analyzed the strongest infrared absorption feature of ethene: the Q-branch of the $\nu_7$ band ($CH_2$ wag) at 949 $cm^{-1}$. This is 7 times stronger than any other feature, including the 3000 $cm^{-1}$ region containing the CH-stretch vibrational modes.

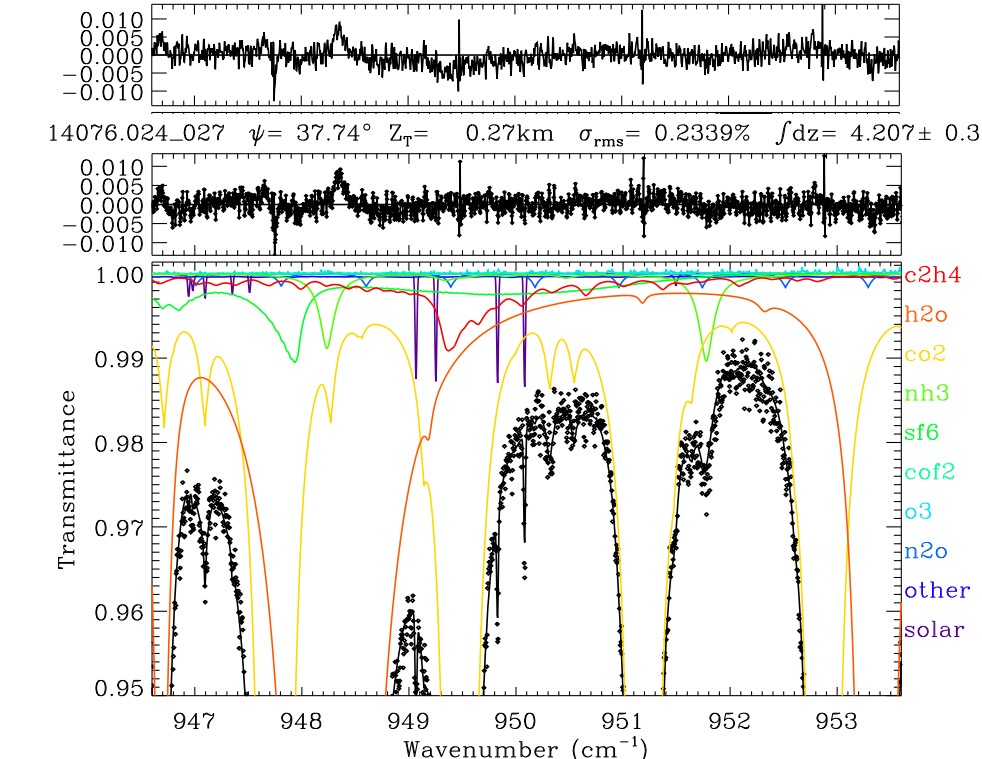

**Figure 2.** *Lower and middle panels are as described in Figure 1, but zoomed in to reveal more detail of the $C_2H_4$ Q-branch (red) whose absorption peaks at 949.35 $cm^{-1}$. The top panel shows residuals from fit performed omitting $C_2H_4$. This causes a discernable 0.5% dip in the residuals around 949.35 $cm^{-1}$ and a worsening of the RMS spectral fits from 0.234% to 0.251%.*

For data acquisition from JPL, the MkIV instrument was indoors with a coelostat mounted to the south wall of the building feeding direct sunlight into the room. Figure 1a shows a fit to the 943-957 $cm^{-1}$ region of one such spectrum. The strongest absorptions are from $H_2O$ lines (orange), one of which is blacked out at 948.25 $cm^{-1}$. There are also eight $CO_2$ lines (amber) in this window with depths of 40-60%, one of which sits directly atop the $C_2H_4$ Q-branch at

949.35 cm$^{-1}$.  These CO$_2$ lines are temperature sensitive, having ground-state energies in the range 1400 to 1600 cm$^{-1}$.  It is not possible to clearly see the C$_2$H$_4$ absorption in Fig. 1, and so Fig. 2 zooms into the Q-branch region.  The lower panel reveals that the peak C$_2$H$_4$ absorption is less than 1% deep and strongly overlapped by CO$_2$.  It is also overlapped by absorption from H$_2$O, SF$_6$, NH$_3$, N$_2$O, and solar OH lines.  NH$_3$ absorption lines exceed 1% in this window on this particular day but do not overlap the strongest part of the C$_2$H$_4$ Q-branch.  The SF$_6$ $\nu_3$ Q-branch at 947.9 cm$^{-1}$ also exceeds 1% but fortunately does not overlap the C$_2$H$_4$ Q-branch.  The SF$_6$ R-branch, however, underlies the C$_2$H$_4$ Q-branch with about 0.3% absorption depth.  The upper panel shows the same spectrum fitted without any C$_2$H$_4$ absorption.  This causes a ~0.5% dip in the residuals around 949.35 cm$^{-1}$ and an increase in the overall rms from 0.234 to 0.251%.  The 0.5% dip in the residuals is weaker than the 0.9% depth of the C$_2$H$_4$ feature in the lower panel because the other fitted gases have adjusted to try to compensate for the missing C$_2$H$_4$.  Their inability to completely do so supports the attribution to C$_2$H$_4$.

Given the severity of the interference, especially the directly-overlying 60%-deep CO$_2$ line, we were at first skeptical that C$_2$H$_4$ could be retrieved to a worthwhile accuracy from this window, or any other. But given the good quality of the spectral fits, we nevertheless went ahead and analyzed the entire MkIV ground-based spectral dataset, consisting of 4379 spectra acquired on 1208 different days over the past 30 years.

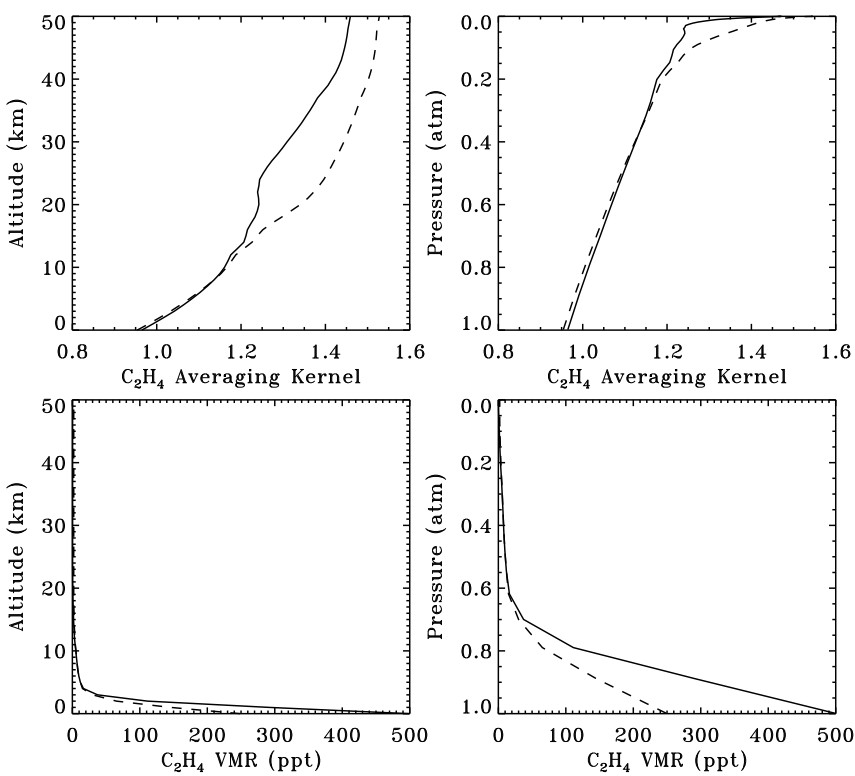

*Figure 3.* *Averaging kernels (upper panels) and a priori volume mixing ratio (vmr) profiles (lower panels) pertaining to the ground-based $C_2H_4$ retrieval illustrated in Figs. 1 and 2. In the left panels quantities are plotted versus altitude. In the right panels, the same data are plotted versus atmospheric pressure. The solid line is the actual profile used. The dashed line is a vmr profile with a less dramatic decrease with altitude: the $C_2H_4$ vmr below 4 km has been halved, with similar amounts in the upper troposphere, and more in the stratosphere. The resulting change in the retrieved total column is only 2%, with the dashed profile giving the lower columns.*

Figure 3 show the averaging kernel and a priori profile pertaining to the $C_2H_4$ retrieval illustrated in Figures 1 and 2. The kernel represents the change in the total retrieved column due to the addition of one $C_2H_4$ molecule.cm$^{-2}$ at a particular altitude. In a perfect column retrieval, the kernel would be 1.0 at all altitudes, but in reality the retrieval is more sensitive to $C_2H_4$ at high altitudes than near the surface, as is typical for a profile-scaling retrieval of a weakly absorbing gas. The a priori vmr profile has a value of 500 ppt at the surface, dropping rapidly to 10 ppt by 5 km altitude. An even larger fractional drop, from 10 to 0.5 ppt occurs in the lower stratosphere between 15 and 21 km. The slight kink in the averaging kernel (solid line) over this same altitude range is due to this large drop in vmr. Since 99% of the $C_2H_4$ lies in the troposphere, the stratospheric portion of the averaging kernel is of academic interest only for total column retrievals.

An important uncertainty in the retrieved column amounts is likely to be the smoothing error, which represents the effect of error in the shape of the a priori vmr profile. If the averaging kernel were perfect (i.e., 1.0 at all altitudes) this wouldn't matter, but in fact the $C_2H_4$ kernels vary from 0.96 at the ground to 1.4 at 40 km altitude. To investigate the sensitivity of the retrieved column to the assumed a priori profile, we also performed retrievals with a different a priori vmr profile in which the $C_2H_4$ vmr profile had been halved in the 0-4 km altitude range and increased in the stratosphere, as depicted by the dashed line in Figure 3. The resulting change in the retrieved $C_2H_4$ column was less than 2% with no discernable change to the rms fitting residuals, which are dominated by the interfering gases. This small $C_2H_4$ column perturbation is a result of the averaging kernel being close to 1.0 at the altitudes with the largest a priori vmr errors (0 to 3 km). Note that only errors in the ***shape*** of the a priori vmr profile affect the retrieved columns in a profile scaling retrieval.

## 3 Results

### 3.1 Ground-based MkIV Retrievals

Figure 4 shows the resulting MkIV ground-based $C_2H_4$ columns from a dozen different observation sites, whose key attributes (e.g. latitude, longitude, altitude, observations, observation days) are presented in the tables of SI.2. The plot is color-coded by the pressure altitude of the site. This was preferred over geometric altitude to prevent all the points from a given site piling up at exactly the same x-value. The pressure altitude varies by up to ±1.5% at the high altitude sites, which is equivalent to ±0.2 km. Only points with $C_2H_4$ uncertainties <1x10$^{15}$ molecules.cm$^{-2}$ were included in the plot, representing 95.7% of the total data volume. One day (out of 258) at Barcroft (3.8 km altitude) was omitted from the plotted data because it had abnormally high $C_2H_4$, as well as other short-lived gases -- clearly a local pollution event.

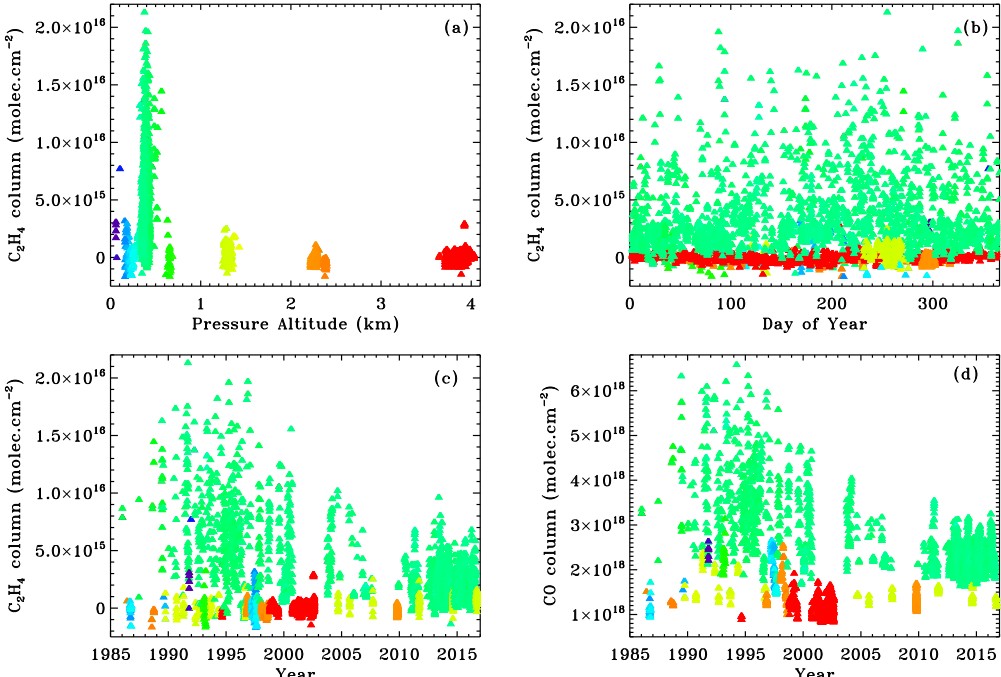

***Figure 4.*** *MkIV column $C_2H_4$ amounts retrieved from 12 different sites, color-coded by pressure altitude. Significant $C_2H_4$ amounts are only found at the urban sites: JPL at 0.35 km altitude (green) and Mountain View at 0.01 km altitude (purple). Panel (b) reveals little seasonal variation in $C_2H_4$. Panel (c) shows a factor 3 decline in $C_2H_4$ in Pasadena over the past 25 years. Panel (d) shows that the CO columns also decreased since 1990, but never come close to zero.*

At all sites above 0.5 km altitude there is essentially no measurable $C_2H_4$. The Table Mountain Facility (TMF) site at 2.3 km altitude (orange) is only 25 km from the most polluted part of the Los Angeles basin, yet no measurable $C_2H_4$ was recorded there in 45 observation days,

despite the good measurement accuracy at this site (see Fig. SI.3). This is probably a result of TMF always being above the PBL (Planetary Boundary Layer), in which urban pollution is trapped, at least on the autumn and winter days when MkIV made measurements at TMF. The high-latitude sites at Fairbanks, Esrange, and McMurdo also have no measurable $C_2H_4$, as do rural, mid-latitude sites (e.g., Ft. Sumner, NM). The only sites where MkIV has ever detected $C_2H_4$ are JPL/Pasadena (0.4 km; green) and Mountain View (0.01 km, purple). These sites are part of major conurbations: Pasadena adjoins Los Angeles; Mountain View adjoins San Jose, California.

The main limitation to the accuracy of $C_2H_4$ measurements by the solar absorption technique is the ability to accurately account for the absorption from $CO_2$, $H_2O$, and $SF_6$, which overlap the Q-branch. The first two gases, in particular, being much stronger absorbers than the $C_2H_4$, have the potential to drastically perturb the $C_2H_4$ retrieval. For example, an error in the assumed $H_2O$ vmr vertical profile, and hence the shape of the $H_2O$ absorption line, will have a large effect on retrieved $C_2H_4$. And since the overlapping $CO_2$ lines are so T-sensitive, a small error in the assumed tropospheric temperature will also greatly influence the $C_2H_4$ retrieval. Errors in the spectroscopy of $H_2O$ and $CO_2$ will also strongly affect $C_2H_4$ retrievals. Figure SI.3 shows the $C_2H_4$ retrieval uncertainties, estimated by solving the matrix equation that relates the jacobians of the various retrieved quantities to the spectral residuals. The uncertainties are the square root of the diagonal elements of the resulting covariance matrix. The same data are plotted versus year, solar zenith angle and site altitude. From JPL the measurement uncertainty is about $0.5 \times 10^{15}$ molec.cm$^{-2}$. At higher solar zenith angles (airmasses) the uncertainty decreases as the $C_2H_4$ absorption deepens. At higher altitudes the uncertainty decreases as the interfering absorptions shrink faster than that of $C_2H_4$. There has been no significant change in the $C_2H_4$ retrieval uncertainty over the 30-year measurement period. We note that the measurements made from McMurdo Antarctica in Sep/Oct 1986 have very small uncertainties, due to their high airmass and the extremely small $H_2O$ absorption. The plotted uncertainties represent a single observation representing a 10-15 minute integration period. 95.7% of the $C_2H_4$ observations have uncertainties $< 1.0 \times 10^{15}$ molec.cm$^{-2}$.

At JPL the $C_2H_4$ column is highly variable. JPL is located at the northern edge of the Los Angeles conurbation, and so when winds are from the Northern sector, or strong from the ocean, pollution levels are much smaller than during stagnant conditions. This is seen in the large range of retrieved $C_2H_4$ values observed at JPL throughout the year. A notable feature of the MkIV $C_2H_4$ data (Fig. 4c) is the factor 3 drop over the past 25 years. In the 1990's $C_2H_4$ often topped $16 \times 10^{15}$ molec.cm$^{-2}$, but since 2010 a column exceeding $8 \times 10^{15}$ has only been observed once.

Figure 4d shows the CO time series at JPL and Fig.5a shows its correlation with $C_2H_4$. CO also shows a substantial decline since the 1990's at JPL, although not as dramatic as that of $C_2H_4$ since CO never falls below $1.5 \times 10^{18}$ molec.cm$^{-2}$ at JPL, even under the cleanest conditions, due to its non-zero background concentration. Figure 5a shows a tight correlation (PCC=0.92) between $C_2H_4$ and CO at JPL (green) suggesting a common local source for both.

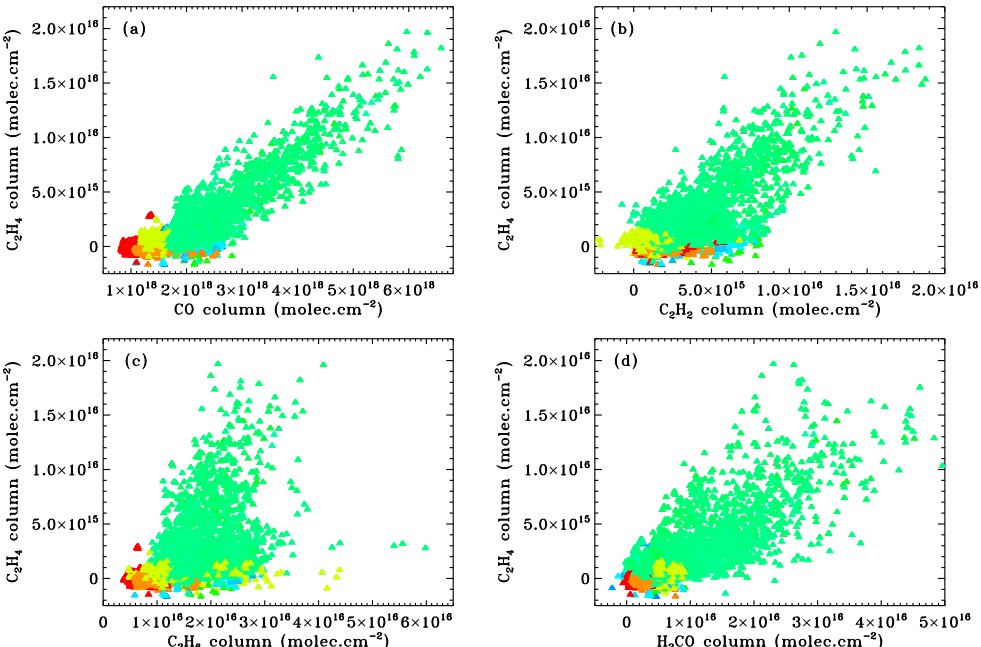

**Figure 5.** *Correlations between $C_2H_4$ and four other gases: (a)=CO; (b)=$C_2H_2$, (c)=$C_2H_6$, and (d)=$H_2CO$. The tightest correlation is with CO with a gradient of 0.0038 and a Pearson correlation coefficient of 0.92 for just the JPL-Pasadena data (green). Poorer correlations exist with the other gases, as low as 0.30 for $H_2CO$. Points color-coded by altitude, as in Figure 4.*

Figure 5b/c/d also shows correlations between $C_2H_4$ and other gases: $C_2H_2$ and $C_2H_6$, and $H_2CO$ for all the MkIV measurements. These correlations are not as tight as that with CO, due to $C_2H_2$ and $C_2H_6$ having other sources. For example, $C_2H_6$ also comes from natural gas leaks. The fact that these trace gases are much less abundant than CO means that their measurements are noisier, which also degrades the correlations.

Figure S.1 plots the gas column relationship for the JPL ground-based data only, each panel containing ~1700 observations. The decreases in the CO, $C_2H_2$, $C_2H_4$ and $H_2CO$ since the 1990's are evident by the lack of red points in the upper right of the panels plotting these gases. $C_2H_6$ has not decreased significantly as is evident from the third row of panels in Fig. SI.1, which shows that the 2015 column abundances (red) span similar values to those measured in 1990 (blue). In fact, on November 10, 2015, we observed a factor 2-3 enhancement of the $C_2H_6$

column as a result of JPL being directly downwind of the Aliso Canyon natural gas leak on that

235     day (Conley et al., 2016).  Although this event was associated with a 2.5% enhancement of

column $CH_4$ (not shown here), there were no enhancements of CO, $C_2H_2$, $C_2H_4$, so these

particular $C_2H_6$ points (red) in the third row of Fig. SI.1 protrude upwards from the main clusters.

| Gas | CO (/ 1000) | $C_2H_2$ | $C_2H_4$ | $C_2H_6$ |
|---|---|---|---|---|
| CO (/ 1000) | | | | |
| $C_2H_2$ | 3.2±0.4   *0.91* | | | |
| $C_2H_4$ | 3.7±0.4   *0.92* | 1.3±0.1   *0.84* | | |
| $C_2H_6$ | 8.9±3.0   *0.45* | 2.4±0.7   *0.61* | 2.6±1.1   *0.30* | |
| $H_2CO$ | 11.1±3.0   *0.57* | 4.9±2.2   *0.44* | 2.7±0.6   *0.67* | -10±14   *-0.04*; |

240

***Table 1.***  *Gradients of the fitted straight lines to the JPL data plotted in Figure SI.1, together with their Pearson correlation coefficients (PCC).  The gradients and their uncertainties are on the left of each cell, the PCC values are italicized on the right.  Note that the CO abundances have*
250     *been divided by 1000 to bring them closer to the other gases.  Thus the Gas:CO gradients are in units of ppt/ppb, whereas the gradients of the non-CO gases are in ppt/ppt.  In the second column, under the header "CO / 1000", the gradients could be termed "emission ratios".*

The highest correlation coefficients are between CO and $C_2H_2$ (0.91) and CO and $C_2H_4$ (0.92).
255     The correlation coefficient between $C_2H_2$ and $C_2H_4$ is only 0.84, probably reflecting the fact that

$C_2H_2$ and $C_2H_4$ are much more difficult  (i.e. noisier) measurements than CO.  The worst

correlation is between $C_2H_6$ and $H_2CO$ (-0.04).

The overall gradient of the $C_2H_4$/CO relationship using all JPL data is 3.7±0.4 ppt/ppb, as

in Table 1, but the post-2010 JPL data have a gradient of only 2.8±0.4 ppt/ppb.  Baker et al.
260     (2008) measured $C_2H_4$/CO emission ratios of 5.7 ppt/ppb in Los Angeles from whole air canister

samples acquired between 1999 and 2005, which is close to their average of all US cities, 4.1

ppt/ppb.  Over this same time period the MkIV JPL data reports 3.9±0.7 ppt/ppb, the larger

uncertainty reflecting the relatively few observations from JPL over this period.  Warneke et al.

(2007) report a $C_2H_4$/CO emissions ratio of 4.9 ppt/ppb in Los Angeles in 2002, measured by
265     aircraft canister samples.  Warneke et al. (2012) report decreases of 6-8% yr[-1] in $C_2H_4$ and CO

over Los Angeles between 2002 and 2010, but little change in the $C_2H_4$/CO emission ratio, which

remained at 5-6 ppt/ppb.

To see whether the ground-based MkIV $C_2H_4$ measured in Pasadena was correlated with the airmass origin, we performed HYSPLIT back-trajectories, and computed the amount of time that airmasses arriving 500 m above JPL had spent over the highly populated areas of Los Angeles conurbation. When column $C_2H_4$ was plotted versus this time-over-conurbation, the correlation was very poor. Column CO also had a poor correlation. The fact that the $C_2H_4$ correlates well with CO tends to discount the possibility that the $C_2H_4$ measurements are wrong, since the CO measurements are very easy. So this implies that the trajectories are not sufficiently accurate. We point out that JPL is located at the foot of the San Gabriel mountains, which rise over 1 km above JPL over a horizontal distance of less than 5 km. This extreme topography might give rise to complexities in the wind fields that might be inadequately represented in the EDAS 40 km-resolution model. Although higher resolution models (e.g. NAM 12km) are available for doing HYSPLIT trajectories, these cover only the past decade, whereas the JPL MkIV measurements go back more than 30 years.

### 3.2 MkIV Balloon Profiles

We also looked for ethene in MkIV balloon spectra using exactly the same window, spectroscopy and fitting software (GFIT) as used for MkIV ground-based measurements. The advantage of the balloon spectra is that the airmass is much larger and the solar and instrumental features are removed from the occultation spectra by ratioing them against a high-Sun spectrum taken at noon from float altitude.

Figure 6 shows a spectral fit to the MkIV balloon spectrum at 6.1 km tangent altitude measured above Esrange Sweden in Dec 1999. The peak $C_2H_4$ absorption at 949.35 cm$^{-1}$ is about 6% deep, although this falls beneath a saturated $CO_2$ line. The information about $C_2H_4$ at this and lower altitudes therefore comes from adjacent weaker features. At higher altitudes (not shown), where the $CO_2$ lines are weaker and narrower, the $C_2H_4$ information comes mainly from the 949.35 cm$^{-1}$ Q-branch.

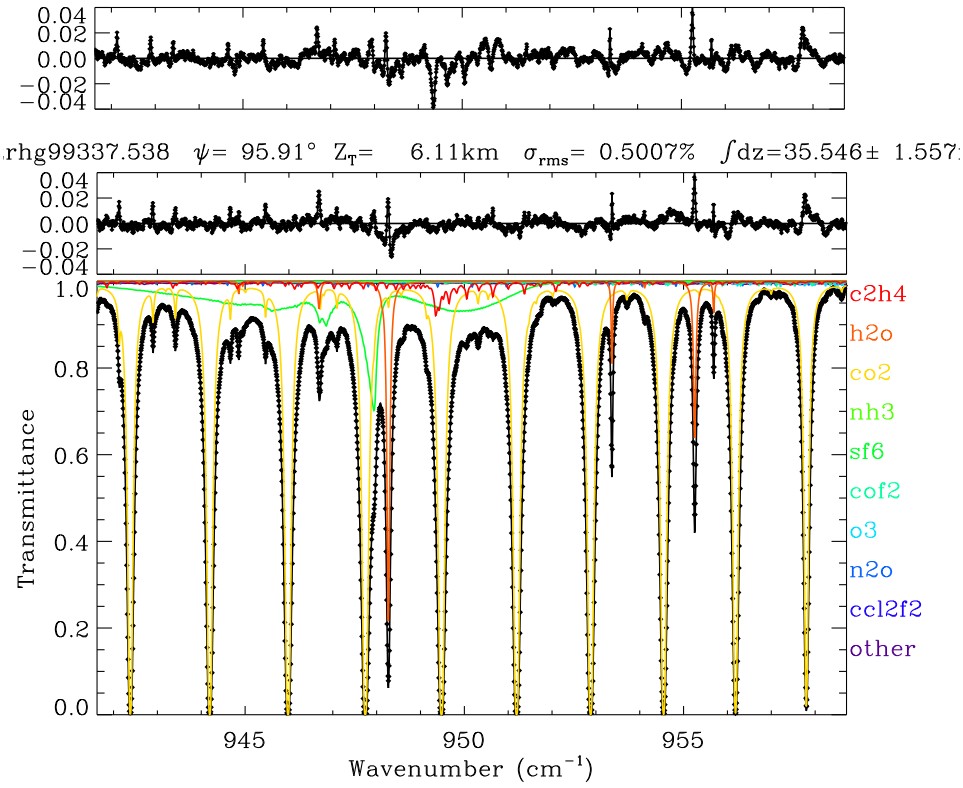

**Figure 6.** *Lower panel shows a fit to a MkIV balloon spectrum with strong $C_2H_4$ absorption measured at 6.1 km tangent altitude. The $C_2H_4$ absorption is denoted by the red line. Its Q-branch is seen at 949.35 $cm^{-1}$ reaching 6% in depth in this particular spectrum. In addition to $C_2H_4$, other gases were adjusted including $H_2O$, $CO_2$, $O_3$, $SF_6$, $COF_2$, $N_2O$, $NH_3$, and $CCl_2F_2$. $CH_3OH$ was included in the calculation but not adjusted. Middle panel shows residuals (measured minus calculated), which are mainly due to $H_2O$. Upper panel shows residuals after omitting $C_2H_4$ from calculation, which causes a large dip in the residuals at 949.35 $cm^{-1}$ and increases the overall rms residual from 0.50% to 0.63%.*

Figure 7 shows 30 balloon profiles of $C_2H_4$ from 23 flights, color-coded according to date. The $C_2H_4$ vmr retrieved from the December 1999 flight (green) was 65±6 ppt at 6 km, decreasing to 14±4 at 7 km, and undetectable above. The remaining balloon flights indicate a 10 ppt upper limit for $C_2H_4$ in the free troposphere and 15 ppt in the stratosphere. Of course, these balloon flights were generally launched under calm, anti-cyclonic, clear-sky conditions, which tend to preclude transport of PBL pollutants up to the free troposphere. So there may be an inherent sampling bias in the MkIV balloon measurements that leads to low $C_2H_4$.

PBL altitudes (0-3 km) are inaccessible from balloon due to the high aerosol content making the long limb path opaque (although they can be probed from the ground). So the balloon measurements are not inconsistent with $C_2H_4$ existing in measurable quantities in the polluted PBL, as implied by ground-based measurements. The typical 1-3 day lifetime of $C_2H_4$ at mid-

and low-latitudes implies that it will only be measureable in the free troposphere soon after rapid
uplift.

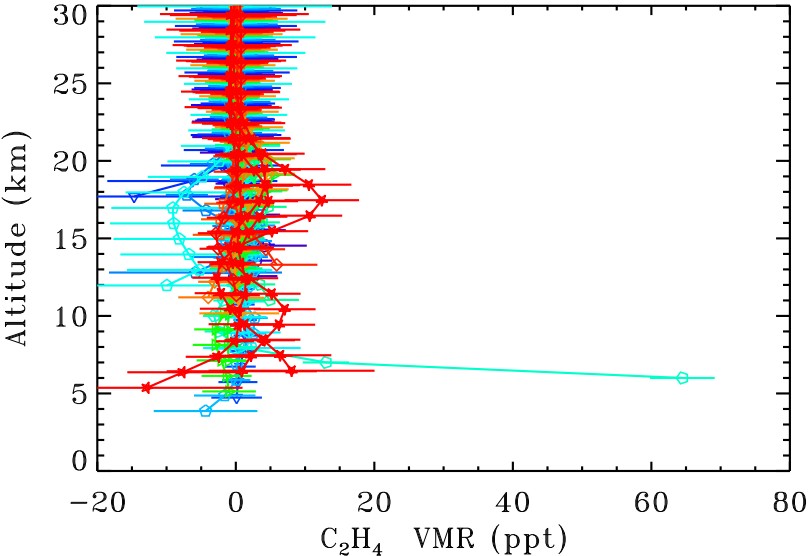

*Figure 7.* *MkIV $C_2H_4$ profiles from 24 balloon flights color-coded by year (purple = 1989; green=2000; red = 2014). Altitude offsets of up to 0.4 km have been applied for clarity, to prevent the error bars from over-writing each other at each integer altitude. In only one flight,*
*launched in Dec 1999 from Esrange, Sweden, was a significant amount of $C_2H_4$ measured (green points at 6-7 km altitude). In other flights there was no detection, with upper limits varying from 10-15 ppt. The increase in uncertainty with altitude above 10 km is due to the $C_2H_4$ absorption feature weakening in comparison with the spectral noise. Below 10 km the increasing uncertainty is due to the greater interference by $H_2O$ and $CO_2$. Note that the negative $C_2H_4$*
*values are all associated with large uncertainties.*

### 3.3 Comparison with Remote Sensing Measurements

Paton-Walsh et al. (2005) measured up to $300 \times 10^{15}$ molec.cm$^{-2}$ of $C_2H_4$ during fire events in SE Australia in 2001-2003 with aerosol optical depths of up to 5.5 at 500 nm wavelength. From spectra acquired during one of the most intense of these fires (Jan 1, 2002), Rinsland et al.
(2005) retrieved a total $C_2H_4$ column of $380 \pm 20 \times 10^{15}$ through a dense smoke plume and inferred a huge mole fraction of 37 ppb peaking at about 1 km above ground level. This retrieval used information from the shape of the Q-branch feature, which was nearly as deep as the overlapping $CO_2$ line. These $C_2H_4$ amounts are 20 times larger than anything seen by MkIV, even from polluted JPL.

Coheur et al. (2007) reported a $C_2H_4$ vmr of $70 \pm 20$ ppt at 11.5 km altitude in a biomass-burning plume, observed by the Atmospheric Chemistry Experiment (ACE) (Bernath et al., 2005) off the East coast of Africa. They also show measured $C_2H_4$ exceeding 100 ppt below 8 km.

Simultaneous measurement of elevated $C_2H_2$, CO, $C_2H_6$, HCN and $HNO_3$ confirm their biomass-burning hypothesis.

Herbin et al. (2009) reported zonal-average ethene profiles above 6 km altitude based on global measurements by ACE. Figure 2 of Herbin et al. shows 35N zonal average vmrs of 40 ppt at 6 km altitude, 30 ppt at 8 km, and 15 ppt at 14 km altitude, with error bars as small as 1 ppt. Herbin et al. (2009) also wrote "We find that a value of 20 ppt is close to the detection threshold at all altitudes in the troposphere". To reconcile these two statements we assume that the 20 ppt

detection limit refers to a single occultation whereas the 1 ppt error bar is the result of co-adding hundreds of ACE profiles.

        Herbin et al. (2009) also report increasing $C_2H_4$ with latitude. Although the ACE zonal means agree with the in situ measurements made during the PEM-West and TRACE-P, these campaigns were designed to measure the outflow of Asian pollution and therefore sampled some

of the worst pollution on the planet. So one would expect lower values in a zonal average. Based on the total absence of negative values in any of their retrieved vmr profiles, we believe that Herbin et al. performed a log(vmr) retrieval, imposing an implicit positivity constraint. This would have led to a noise-dependent, high bias in their retrieved profiles in places where $C_2H_4$ was undetectable.

Clerbeaux et al. (2009) reported $C_2H_4$ column abundances reaching $3 \times 10^{15}$ molec.cm$^{-2}$ from spectra acquired by the IASI satellite instrument, a nadir-viewing emission sounder. This isolated event occurred on May 2008 over Eastern Asia and was associated with a Siberian fire plume, as confirmed by back-trajectories and co-located enhancements of $CH_3OH$, HCOOH and $NH_3$.

More recently, $C_2H_4$ was detected in boreal fire plumes (Alvarado et al., 2011; Dolan et al., 2016) during the 2008 ARCTAS mission by the Tropospheric Emission Sounder (TES), a nadir-viewing thermal emission FTS on board the Aura satellite. A strong correlation with CO was observed. TES's $C_2H_4$ sensitivity depends strongly on the thermal contrast: the temperature of the $C_2H_4$ relative to that of the underlying surface. For plumes in the free troposphere a detection

limit of 2-3 ppb is claimed from a single sounding with a 5 x 8 km footprint.

### 3.4 Comparison with In Situ Measurements

        There are a lot of published in situ ethene measurements. Here we intend to discuss only those that are in some way comparable with MkIV measurements. These include measurements

over the Western US and profiles over the Pacific Ocean in the 30-40ºN latitude range that are

upwind of MkIV balloon measurements.  Other measurements, e.g., over Europe and mainland SE Asia, are less relevant, given the 1-3 day lifetime of $C_2H_4$.

Gaffney et al. (2012) reported surface $C_2H_4$ over Texas and neighboring states measured in 2002.  They reported a median vmr of 112 ppt, with occasional much larger values of up to 2
ppb, presumably when downwind of local sources.  This median value, if present only within a 150 mbar-thick PBL, represents a total column of $0.3 \times 10^{15}$ molec.cm$^{-2}$, which would be undetectable in ground-based MkIV measurements.

Lewis et al. (2013) reported airborne in situ measurements of non-methane organic compounds over SE Canada in summer 2010.  The median ethene vmr was 49 ppt with plumes
averaging 1848 ppt.  Their $C_2H_4$/CO scatter plot (Fig. 2b of Lewis et al.) reveals two distinct branches.  Biomass burning plumes show an emission ratio of 6.97 ppt/ppb, whereas "local/anthropogenic emissions" show an emission ratio of about 1.3 ppt/ppb.  These values bracket the MkIV value of 3.7±0.5 ppt/ppb obtained from the green points in Fig. 5a and all points of Fig.SI.1 of the current paper.

Blake et al. (2003) report mean $C_2H_4$ profiles from 0 to 12 km during the Feb-Apr 2001 TRACE-P aircraft campaign, during which aircraft based in Hong Kong and Tokyo sampled outflow from SE Asia.  Blake et al. compared these results with those from the similar 1991 and 1994 PEM-West campaigns.  Blake et al.'s Figure 11 shows that below 2 km $C_2H_4$ averaged 100 ppt during TRACE P and 250 ppt during PEM-West.  Blake et al.'s Table 1 provides a median
$C_2H_4$ of 30 ppt at 35N at 2-8 km altitude in the Western Pacific for both TRACE-P and PEM-West.  Below 2 km the vmrs were much larger, especially during PEM-West.  Blake et al.'s Figure 9 shows mean PBL vmrs of 200 ppt during Trace-P and 400 ppt during PEM-West, rapidly decreasing to 50 ppt by 4 km altitude, 30 ppt by 6 km, and less than 20 ppt above 9 km.  Blake et al.'s Figure 2 shows high $C_2H_4$ in the coastal margins of China, decreasing rapidly by a
few hundred km off shore, consistent with the short $C_2H_4$ lifetime.  Since these aircraft campaigns were designed to measure polluted outflow from East Asia, their samples cannot be considered representative of a zonal average. Over the mid-Pacific, $C_2H_4$ amounts were 0-15 ppt at all altitudes during TRACE-P and PEM-West B.

Sather and Cavender (2016) reported surface in situ measurements of ozone and Volatile
Organic Compounds (including ethene) from the cities of Dallas-Ft. Worth, Houston, El Paso, Texas, and from Baton Rouge, Louisiana, over the past 30 years.  For ethene the measurements span the late 1990s to 2015, but nevertheless show clear declines by factors of 2-4 during 5-8am on weekdays.  The authors attribute this decrease to the impacts of the 1990 amendment to the US Clean Air Act.

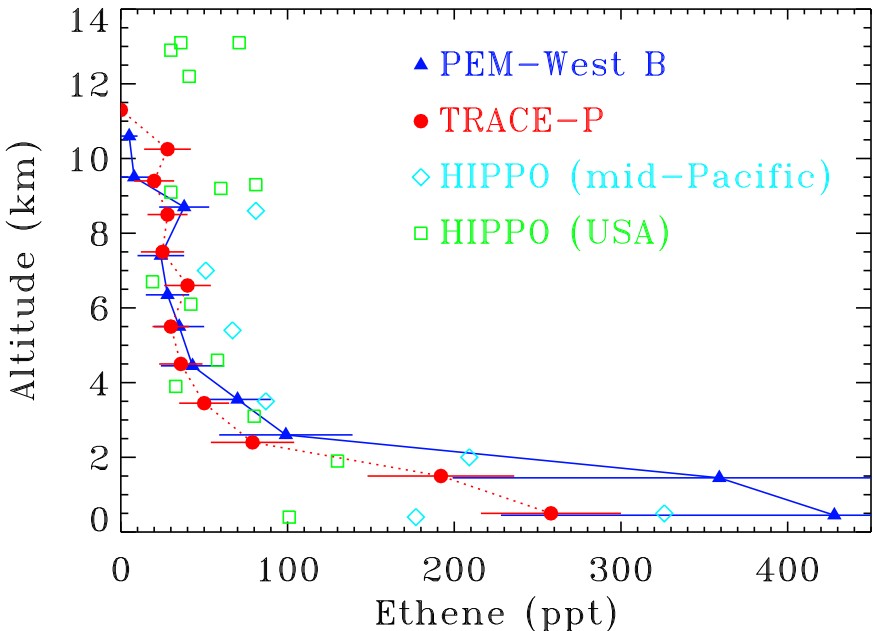


**Figure 8.** *Aircraft in situ measurements of ethene. HIPPO measurements of $C_2H_4$ made by the Advanced Whole Air Sampler between 30° to 40°N are shown by cyan diamonds (mid-Pacific) and green squares (Central USA). Also shown are PEM-West B (blue triangles) and TRACE-P (red circles) measurements of $C_2H_4$ over coastal SE Asia and the Western Pacific (taken from Figure 9 of Blake et al. (2003)).*


Ethene was measured during the HIAPER Pole-to-Pole (HIPPO; Wofsy et al. 2011, 2012) mission by the Advanced Whole Air Sampler. Figure 8 plots the $C_2H_4$ vmrs measured in the 30-40N latitude range. Points are color-coded by longitude. The cyan points were measured

mid-Pacific in Jan and Dec 2009, Apr 2010, and Jun/Jul 2011. The green points were measured over the Central/Western USA in Jan and Dec 2009, and Jun/Jul 2011. Profiles from the PEM-West B and TRACE-C aircraft campaigns are plotted in red and blue. Surprisingly, $C_2H_4$ is larger over the mid-Pacific (blue/purple points) than the USA (red points) at altitudes below 9 km. This is presumably due to Asian pollution being further destroyed while crossing the Eastern

Pacific. Above 9 km the $C_2H_4$ is larger over the USA, presumably due to upward transport of the Asian pollution.

Washenfelder et al. (2011) performed ground-based in situ measurements from Pasadena, California, of several glyoxl precursors in early June 2010, as part of the CalNEX 2010 campaign. An ethene mole fraction of 2.16 ppb was reported. Assuming that this value was

present throughout the PBL, extending from the surface at 1000 mbar to the 900 mbar level, then the in situ measurement implies a total $C_2H_4$ column of $4\times10^{15}$ molec.cm$^{-2}$, which is consistent

with the upper range of values observed by MkIV in 2010.  Unfortunately we do not have temporally overlapping measurements, and even if we did JPL is 10 km from the Pasadena site.

Washenfelder et al. (2011) also report a factor 6 drop in $C_2H_4$ amounts since the September 1993 CalNEX campaign, but note that the 1993 readings occurred during a smog episode, implying higher than normal levels of pollution.  This drop is larger than the factor 3 decrease seen in the MkIV column data, but not inconsistent given the sparse statistics together with the large day-to-day variability seen in the MkIV data.

Measurements of ethene from ground level in Mexico City in 1999, 2002, and 2003 ranged between 10-60 ppb, with higher levels in the commercial sectors and lower values in residential areas (Altuzar et al., 2001, 2005; Velasco et al., 2007).  These are 5-30 times larger than the 2.16 ppb measured by Washenfelder et al. (2011) in Pasadena in 2010.

## 4  Discussion

The simultaneous reductions in CO and $C_2H_4$ ground-based column amounts measured
from JPL over the past 25 years, and their continued high correlation, suggest a common source: vehicle exhaust.  The declines in CO and $C_2H_4$ are likely a result of improved vehicle emission control systems, mandated by the increasingly stringent requirements imposed by the US Environmental Protection Agency (EPA; e.g., the 1990 Clean Air Act), various state laws, and the California Air Resources Board (CARB, LEV2) over the past decades and stronger
enforcement thereof (e.g., smog checks).  This view is supported by Bishop and Stedman (2008) who showed that vehicle emissions of hydrocarbons in several US cities including Los Angeles have steadily decreased with vehicle model year since 1986.

$C_2H_4$/CO emission ratios measured over JPL by MkIV have decreased over the 30 year record, from 3.7±0.4 ppt/ppb overall to 2.7±0.4 ppt/ppb in recent years.  It is not clear what is
causing this decrease since many things have changed that might affect $C_2H_4$ levels (e.g. regulation of internal combustion engine exhaust, elimination of oil-based paints and lighter fuel, better control of emissions from oil and gas wells).

MkIV balloon measurements have only detected ethane once in 24 flights: in the Arctic in December 1999 at altitudes below 6 km.  In all other flights an upper limit of 15 ppt was
established for the free troposphere and 10 ppt for the lower stratosphere.  These upper limits are substantially smaller than the ACE 35N zonal mean profiles reported by Herbin et al. (2009), which are possibly biased high when $C_2H_4$ amounts are small due to a positivity constraint imposed on the retrievals.  Also, a single biomass burning plume with up to 25 ppb of $C_2H_4$ has the potential to significantly increase the zonal mean $C_2H_4$.  For this reason, a zonal median

470 would be a more robust statistic.  It is also possible that the MkIV balloon flights under-represent conditions in which PBL pollution is lofted due to their location and the meteorology associated with balloon launches.  Herbin et al. (2009) reported an increase of the 6-km ACE $C_2H_4$ with latitude in the Northern hemisphere, peaking at 53 ppt at 70°N.  This is consistent with the December 1999 MkIV balloon flight from 67°N, which measured 60 ppt at 6 km.

475   MkIV balloon measurements over the Western USA reveal much smaller ethene amounts than in situ aircraft measurements over SE Asia during TRACE-P, PEM-West B, and over the mid-Pacific ocean during HIPPO.  With its 1-3 day lifetime, $C_2H_4$ decreases substantially during its Eastward journey across the Pacific, which would help reconcile them with the MkIV balloon profiles.


## 5  Summary and Conclusions

   A 30-year record of atmospheric $C_2H_4$ has been extracted from ground-based FTIR spectra measured by the JPL MkIV instrument.  Despite its high sensitivity, MkIV only detects ethene at polluted urban sites (e.g., Pasadena, California).  At clean sites visited by MkIV, $C_2H_4$

485 was undetectable (less than $10^{15}$ molec.cm$^{-2}$).  MkIV ground-based measurements are generally consistent with the available surface in situ measurements, although a definitive comparison is difficult due to the large variability of $C_2H_4$ and lack of co-incidence.

   A large decline in $C_2H_4$ has been observed over Pasadena over the past 25 years.  This is likely the result of increasingly stringent requirements on vehicle emissions imposed by the US

490 Environmental Protection Agency (e.g., the 1990 Clean Air Act) and the California Air Resources Board (Low Emission Vehicle 2 requirements) over the past decades, together with stronger enforcement of these regulations (e.g., smog checks).  The $C_2H_4$/CO emissions ratio also appears to have decreased in recent years.

   This work shows that $C_2H_4$ might in future become a routine product of the NDACC

495 Infra-Red FTS network, at least at sites with local sources.  Moreover, since the spectra are saved, a historical $C_2H_4$ record may be retro-actively extractable at some of the more polluted sites.

**Acknowledgements.**  This research was performed at the Jet Propulsion Laboratory, California Institute of Technology, under contract with the National Aeronautics and Space Administration.
500 We thank the Columbia Scientific Balloon Facility (CSBF) who conducted the majority of the balloon flights.  We also thank the CNES Balloon Launch facility who conducted two MkIV balloon flights from Esrange, Sweden.  We thank the Swedish Space Corporation for their support and our use of their facilities.  We thank the HIAPER Pole-to-Pole Observations (HIPPO) campaign for use of their data.  Finally, we acknowledge support from the NASA Upper
505 Atmosphere Research Program.  © 2017 California Institute of Technology. Government sponsorship acknowledged.

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

**Figure Captions**

*Figure 1. Example of a fit to a ground-based MkIV spectrum measured from JPL, California, on March 17, 2014 at a solar zenith angle of $\psi = 37.7°$ from a pressure altitude of $Z_T = 0.27$ km. In the lower panel black diamond symbols represent the measured spectrum, the black line represents the fitted calculation, and the colored lines represent the contributions of the various absorbing gases; mainly $CO_2$ (amber) and $H_2O$ (orange). Also fitted are the 0% and 100% signal levels, separate telluric and solar frequency shifts, together with 5 more weakly absorbing gases ($NH_3$, $SF_6$, $COF_2$, $O_3$ and $N_2O$). The retrieved $C_2H_4$ column amount on this day, $4.2 \times 10^{15}$ molec.cm$^{-2}$, would represent 2 ppb confined to the lowest 100 mbar (1.5 km) of the atmosphere. The $C_2H_4$ absorption contribution (red) peaks at 949.35 cm$^{-1}$ with an amplitude of less than 1% and therefore difficult to discern on this plot. The upper panel shows fitting residuals (measured - calculated), peaking at 1.3% with an rms deviation of 0.234%, which are mainly correlated with $H_2O$ and $CO_2$. The vertical column of $C_2H_4$ derived from this fit was $4.2 \pm 0.4 \times 10^{15}$ molec.cm$^{-2}$.*

***Figure 2.***  *Lower and middle panels are as described in Figure 1, but zoomed in to reveal more detail of the $C_2H_4$ Q-branch (red) whose absorption peaks at 949.35 $cm^{-1}$.  The top panel shows residuals from fit performed omitting $C_2H_4$.  This causes a discernable 0.5% dip in the residuals around 949.35 $cm^{-1}$ and a worsening of the RMS spectral fits from 0.234% to 0.251%.*

***Figure 3.*** *Averaging kernels (upper panels) and a priori volume mixing ratio (vmr) profiles (lower panels) pertaining to the ground-based $C_2H_4$ retrieval illustrated in Figs. 1 and 2.  In the left panels quantities are plotted versus altitude. In the right panels, the same data are plotted versus atmospheric pressure. The solid line is the actual profile used. The dashed line is a vmr profile with a less dramatic decrease with altitude: the $C_2H_4$ vmr below 4 km has been halved,*
*with similar amounts in the upper troposphere, and more in the stratosphere. The resulting change in the retrieved total column is only 2%, with the dashed profile giving the lower columns.*

       ***Figure 4.***  *MkIV column $C_2H_4$ amounts retrieved from 12 different sites, color-coded by pressure altitude.  Significant $C_2H_4$ amounts are only found at the urban sites: JPL at 0.35 km altitude*
*(green) and Mountain View at 0.01 km altitude (purple).  Panel (b) reveals little seasonal variation in $C_2H_4$.  Panel (c) shows a factor 3 decline in $C_2H_4$ in Pasadena over the past 25 years.  Panel (d) shows that the CO columns also decreased since 1990, but never come close to zero.*

       ***Figure 5.***  *Correlations between $C_2H_4$ and four other gases: (a)=CO; (b)=$C_2H_2$, (c)=$C_2H_6$, and*
*(d)=$H_2CO$.  The tightest correlation is with CO with a gradient of 0.0038 and a Pearson correlation coefficient of 0.92 for just the JPL-Pasadena data (green).  Poorer correlations exist with the other gases, as low as 0.30 for $H_2CO$.  Points color-coded by altitude, as in Figure 4.*

       ***Figure 6.***  *Lower panel shows a fit to a MkIV balloon spectrum with strong $C_2H_4$ absorption*
*measured at 6.1 km tangent altitude.  The $C_2H_4$ absorption is denoted by the red line.  Its Q-branch is seen at 949.35 $cm^{-1}$ reaching 6% in depth in this particular spectrum.  In addition to $C_2H_4$, other gases were adjusted including $H_2O$, $CO_2$, $O_3$, $SF_6$, $COF_2$, $N_2O$, $NH_3$, and $CCl_2F_2$.  $CH_3OH$ was included in the calculation but not adjusted.  Middle panel shows residuals (measured minus calculated), which are mainly due to $H_2O$.  Upper panel shows residuals after*
*omitting $C_2H_4$ from calculation, which causes a large dip in the residuals at 949.35 $cm^{-1}$ and increases the overall rms residual from 0.50% to 0.63%.*

       ***Figure 7.***  *MkIV $C_2H_4$ profiles from 24 balloon flights color-coded by year (purple = 1989; green=2000; red = 2014). Altitude offsets of up to 0.4 km have been applied for clarity, to*
*prevent the error bars from over-writing each other at each integer altitude.  In only one flight, launched in Dec 1999 from Esrange, Sweden, was a significant amount of $C_2H_4$ measured (green points at 6-7 km altitude).  In other flights there was no detection, with upper limits varying from 10-15 ppt.  The increase in uncertainty with altitude above 10 km is due to the $C_2H_4$ absorption feature weakening in comparison with the spectral noise.  Below 10 km the increasing*
*uncertainty is due to the greater interference by $H_2O$ and $CO_2$.  Note that the negative $C_2H_4$ values are all associated with large uncertainties.*

**Figure 8.** *Aircraft in situ measurements of ethene. HIPPO measurements of $C_2H_4$ made by the Advanced Whole Air Sampler between 30° to 40°N are shown by cyan diamonds (mid-Pacific) and green squares (Central USA). Also shown are PEM-West B (blue triangles) and TRACE-P (red circles) measurements of $C_2H_4$ over coastal SE Asia and the Western Pacific (taken from Figure 9 of Blake et al. (2003)).*

## Table

**Table 1.** *Gradients of the fitted straight lines to the JPL data plotted in Figure SI.1, together with their Pearson correlation coefficients (PCC). The gradients and their uncertainties are on the left of each cell, the PCC values are italicized on the right. Note that the CO abundances have been divided by 1000 to bring them closer to the other gases. Thus the Gas:CO gradients are in units of ppt/ppb, whereas the gradients of the non-CO gases are in ppt/ppt. In the second column, under the header "CO / 1000", the gradients could be termed "emission ratios".*

## Supplementary Information

**Figure SI.1.** *Relationships between MkIV gas column abundances measured from JPL, color-coded by year (blue = 1990; green = 2000; orange = 2010; red = 2015). Panels on the same row all have the same y-axis, avoiding having to repeat the y-annotation. Panels in the same column have the same x-axis, avoiding repeating the x-annotation. Correlation coefficients pertaining to these panels can be found in Table 1 of the main paper.*

**Table SI.2a.** *Summarizing the twelve observation sites from which the JPL MkIV instrument has made ground-based observations as of the end of 2016, sorted by latitude. For each site, the latitude, longitude, and altitude are listed, together with the type of surrounding terrain, the season, and time of day when observations were typically made. The number of observations ($N_{obs}$) and observation days ($N_{day}$) from each site are also provided. JPL has the most observation days with 648+5=653, followed by Mt. Barcroft (258), and Ft. Sumner (89).*

**Table SI.2b.** *MkIV ground-based observation days at the 12 different sites, broken down by year. 2001 was the year with the most observations days (101) all from Mt. Barcroft. JPL is the site with the most observation days (653).*

**Figure SI.3.** *$C_2H_4$ retrieval uncertainties, color-coded by pressure altitude, plotted versus year (top), solar zenith angle (middle), and pressure altitude (bottom). At any given site, ethene uncertainties decrease with solar zenith angle as the absorption features deepen (middle panel). In absolute terms the uncertainties are smallest at the highest altitude sites (lower panel), where ethene is virtually never detectable. In fractional terms, ethene uncertainties are smallest at the low altitude polluted sites.*