# Peer review of "Measurements of atmospheric ethene by solar absorption FTIR spectrometry"

_Atmospheric Chemistry and Physics, 2017_

## Referee Comment (RC1) · Anonymous Referee #1 · 3 Jul 2017

This paper describes the retrieval of atmospheric ethene amounts from 30 years of solar remote sensing measurements from the ground and stratospheric balloon. Ethene absorbs only weakly in the spectra, and the analysis is very careful, comprehensive and reliable. The measurements are described in detail, and compared to a range of other published remote sesnig and in situ measurements. The conclusions focus on the observed decrease in C2H4 amounts over the period, but the discussion of reasons for this decline is somewhat disjointed and anecdotal. A restructure of the discussion and conclusions sections would improve the paper and do justice to the high quality measurements.

The paper is suited to ACP readers and I recommend publication after minor revisions listed below.

[Figure]

L 11 ethane should read ethene

L 11 1990s not 1990's

L 17-18. Suggest rephrasing without "etc." for example "Atmospheric ethene is formed primarily by incomplete combustion from sources such as biomass burning, power plants and combustion engines."

L 17-25 Ambiguity: Sawada and Totsuka are referenced 3 times, redundantly, together with Goldstein, but it is not clear if the referenced fluxes are all fluxes or just biogenic. Please rephrase.

L 31 Not all readers will be familiar with "MkIV" - briefly describe "MkIV" at first use, for example "... with the remote sensing results described here from the MkIV solar infrared interferometer.

L44 The website reference may change in the future. Could a snapshot or table of the webpage be added as an appendix or supplementary info to the paper?

L 55/Fig 1 The red $C_2H_4$ and $H_2O$ spectra are almost impossible to distinguish, please choose a colour that makes $C_2H_4$ stand out. It is in fact clearer in Figure 2, but the reader may not realise this.

L107 This text is exactly repeated in the Figure 2 caption. It could be left out of one or the other.

L125 The solid lines in Figure 3 show... (not shows...)

L 148 The names and locations of the 12 sites should be provided here or in a table. Many (most) readers will not be familiar with the MkIV sites.

L177 "Where" or "While" rather than "Whereas"?

L 184 et seq. This statement is out of place here – it is more in the nature of discussion than presentation of results, and represents the authors' opinion without real quantitative backup. This sentence and other similar points in the results section may be better collected in the discussion section.

L198 have => having

Figure 6 The colours are different from those in Fig 1 & 2, can they be made consistent (with C2H4 easy to distinguish from other gases).

L 296 spell out AOD and MIR

L311 Figure 2 of ???? et al.

L 319 Is it possible to ask Herbin or co-author rather than speculate about the log-retrieval?

L321 retrieval mis-spelt

L335 Ethene measurements, not ethane

L337 lower case P in Profiles

Discussion L404-417. This paragraph is a little frustrating – do the authors believe that the urban areas are NOT responsible for the high CO and C2H4, or that the trajectories are inaccurate ("wrong" is a strong word). Having mentioned both possibilities, the discussion is left hanging without any conclusion.

L418-424 There a few points scattered through the results pointing towards the secular decrease and its causes that could be pulled together here into the discussion, which is currently rather light. Some of this discussion currently resides in the summary and conclusions – I would recommend expanding the discussion and reducing the Conclusions to the main points of the study

---

## Referee Comment (RC2) · Anonymous Referee #2 · 8 Jul 2017

This paper reports measurements of atmospheric ethene columns at a dozen sites. Downward trends were observed in Pasadena, likely related to regulations on vehicle emissions. This paper presents interesting information and should be published, but much of the discussion is too speculative and needs to be strengthened. Basic information about the sampling sites/times and instrument performance is needed. The impact of an apparently high detection limit on the results needs to be clarified, for example is lack of seasonality or "essentially no" C2H4 more related to the instrument rather than actual lack of seasonality? The presentation also needs some clarification and restructuring to make it easier for the reader to follow.

General comments:

Abstract: The abstract could be filled out by stating roughly where the dozen sites

are located, how many sites were clean-air vs urban, how many sites fell below the detection limit, and how many sites showed measurable trends. Currently only 4 clean air sites and one urban site are discussed.

General comment: There is a lot of colloquial language in the paper (L56, L65, L90, L335, L393, etc.). Use specific scientific language.

Introduction: The first sentence seems strange because it misses the biogenic source. What is the partitioning of combustive vs biogenic ethene in its global budget? If the biogenic source is 21 Tg/yr, how large is the combustive source? Were Poisson et al. measuring biogenic ethene? It's not in the references. Overall the first paragraph needs tightening up.

L65: "This is a "greatest hits" compilation . . . not always the latest version for every band of every gas". This is too vague . . . be more specific in describing the limitations and give some sense of your overall precision and how this varied over the years or at different sites.

L95: Before getting into specific Pasadena results we need a Table stating the 12 sites, their latitude, longitude and altitudes, and the dates over which measurements were taken, what time of day and for how long, whether various sites were considered clean or polluted, etc. For example the "Ground-based Sites and Observations (1985-2015)" table from http://mark4sun.jpl.nasa.gov/ground.html (L44) could be included, with sample number and dates added in (observation days are only listed until 2004 on the website).

L152: We still don't know basics such as the precision of the instrument (I'm assuming the detection limit is 10ˆ15 molec.cm2 based on the abstract, but it's not clearly stated). What is meant by "uncertainties" here, and what is the basis for choosing <1x10ˆ15?

L160: The finding of little seasonal variation is surprising, e.g. Herbin et al. (2009) saw clear seasonal ethene variations in both hemispheres. Is ethene's seasonal variation

below the detection limit, especially at clean sites? What detection limit would you need to detect seasonality, for example in Sweden (high northern latitude) in winter? On L164 ("there is essentially no C2H4") use more precise language. What exactly can you measure?

L162: "but never come close to zero". Why is this stated? CO has a non-zero background concentration, so it wouldn't be expected to approach zero. On L181 the statement that the CO decline is not as dramatic as ethene because it doesn't fall as low doesn't make sense for the same reason. By what factor did CO fall relative to its background, and how does that compare to ethene? What is the clean background value for CO at Pasadena latitudes in winter and summer?

L169: Where is the TMF site relative to LA (what direction)? Were winds from a clean or polluted direction? On L168 this is the first time we're told what season the measurements were made.

Figure 4 needs a legend to match color with site . . . which site goes with which altitude? Same with Figure 5. Though altitude may not be the best way to color-code . . . places as different as McMurdo and Texas come out as the same color and the 12 sites can't be distinguished.

L170: "The only sites where MkIV has ever detected C2H4. . ." What color is Sweden and what season was it measured (winter)? There seems to be some light blue around 1989 that is measurable? In each panel of Figure 4, a line should be drawn indicating the detection limit.

Line 175: The evidence for this is a little weak . . . a wind rose plot would clearly show how the ethene levels vary with wind direction and how much of the large range is related to wind direction versus other things like time of day or seasonality.

L183: The arguments in this paragraph, while most likely correct, are too speculative. Apart from the Clean Air Act of 1990, when did CARB policies and stronger enforcement of smog checks occur? Is the ratio of C2H4/CO consistent with traffic? If you believe it's traffic, why is biomass burning raised as a possibility on L422-423? It may help to discuss the Sather and Cavender and Washenfelder et al. papers here rather than below.

L196: What do you mean by "not as tight"? What is the r2? What other sources do you expect ethyne to have in an urban center? Unlike ethane it's also a combustion tracer like ethene, so it should behave like CO and ethene. What does the correlation between CO and ethyne look like?

L205: Why "seems"? What does statistical analysis show? Do not use wording like "large values of the red points in the third row". Use scientific descriptions.

L230: Since the ratio appears to be changing over time, what is you ratio for 1999-2005, the same time-frame as Baker et al.? That would be better than comparing 1985-2016 to 1999-2005.

L334: Suggest condensing this section. It's sort of a laundry list of other projects without much synthesis. On L338 if SE Asian measurements are not relevant to this study, why are they presented (e.g., Blake et al., 2003; Figure 8)? Probably a Table would be a better way to intercompare results and show the different years, seasons and locations of each campaign.

L381: In comparing the mid-Pacific to the USA, you need to state what year and season the different missions flew and what impact this might have had (the figure is comparing winter/spring flights from 1994-1999 with HIPPO over a decade later). What phase of HIPPO is plotted? Are the mid-Pacific and USA HIPPO data from the same season? Is L381-382 referring only to the HIPPO data (which is blue and green in the figure; no red points)? On L384 what evidence was there for upward transport of Asian pollution to high altitude? Is this referring to HIPPO data? Overall this paragraph needs tightening and better links to the rest of the paper.

L409: Use less speculative arguments . . . "tends to discount the possibility that the C2H4 measurements are wrong" is not convincing. Suggesting that the trajectories are wrong or that urban pollution isn't a major source of ethene or CO probably isn't the direction you want to go. Do you get better correlations when you remove data originating from the San Gabriel direction? L418-423 is too speculative. No evidence was provided for decreasing emissions from biomass burning in the Pasadena area . . . if you believe this is the case your argument needs to be much more substantial.

Minor corrections/clarifications:

General: I suggest numbering your sections so they fall more clearly into Methods, Results etc.

L7: All acronyms need to be defined, even if they seem obvious. Define JPL. On L22 define PAR, and so forth (L98, L120, L220, L296).

L11: Ethane should be ethene. Same on L335 and L346.

L50: I don't see a black line in the lower panel.

L52: Many of these terms (continuum level, etc.) might not mean much to the average reader.

L56: "is less than 1% deep" . . . use more specific wording or define deep.

L57: 0.235% is a very precise number . . . is this the level of significance you intend?

L58: Use spaces to indicate minus rather than a hyphen in "measured-calculated".

L77: Is this what you used? Just this sentence is out of place without some link to your study.

L95: Give an exact lat/long and describe the site. In a field? Near a road? Wind direction?

L102: State which day. The caption just says March 2014.

L107-110: This is almost the same wording as the caption.

L125-126: Similar wording as the caption.

L131: Text uses ppt. Is the x-axis of the lower panels also ppt? Units aren't given and the text is very difficult to read because of scientific notation. Just use ppt and a scale from 0 to 500.

L136-148: This is more Methods than Results.

L144: "smallness" is a strange word. How about just "This small $C_2H_4$ column perturbation…"

L152: Units are needed for $1x10^{15}$.

L164: Define TMF. This site designation doesn't mean anything to the reader.

L171: What was the season, time of day and wind direction for the Mountain View measurements?

L173: No capital for Northern.

L181: Change "The CO" to "CO" or "The CO column". Same on L203.

L183: State the r2 value after "tight correlation".

L193: "poorer" … how poor compared to 0.93? How about Mountain View, the other urban site?

L198: Should be having not have.

L204: Awkward wording.

L208: 2016 not 2916.

L208: Typo … "a 2.5% enhancements". Similar issue on L243.

L224: An emission ratio subtracts off the background. Was that done here for CO?

L229: What is the distinction between JPL ground-based data only (Table 1) and "all JPL data"?

L231: Did Baker et al. and Warneke et al. report uncertainties?

L263: Why 949.4 here but 949.35 on L246?

L272: Use consistent units. The text cites 65 ppt but Figure 7's x-axis uses 6x10-11. Just use ppt.

L297-298: Not sure of the point of this statement relative to your paper.

L305: Not necessary to include "their Table 2" and "Their Fig. 2".

L311: Correct "of et al." The paragraph needs some re-writing . . . it's too casual.

L314: Why "presumably"? Is it not clear from the paper? Same on L320 and L342.

L321: Typo, retrieval.

L335: Too colloquial. This paragraph needs proofing.

L347: 1848 is too precise for an average; add an error bar if they had one. Same for on L348 for the ER of 6.97.

L351: Typo, "fig.S1".

L368: No comma after measurements.

L377: HIAPER not HAIPER.

L381: I think you mean "green squares" rather than "red points". Same on L382.

L386, L394: No comma after al.

L387: No hyphen for precursor.

L399: When were the measurements in Mexico City? Are they ground-level?

L404: This is methods/results more than discussion.

The conclusions read more like a summary.

References need consistent formatting.

Table 1: The significant figures in the gradient and error need to match: 1.3 +/- 0.1 but not 1.28 +/- 0.1 and so forth.

Figure 1: Tidy up the graph for publication (stronger font, less writing on top – or if you include it define all the symbols). In Figures 2 and 6 the writing on top is cut off.

Figures 1 and 2: Is the top panel in Figure 1 the same as the middle panel in Figure 2? If so delete the top panel in Figure 1. The two figures could probably be merged.

Figure 3: Axis labels are not clear (are overlaid). Need a larger font.

Figure 5 could be on a log scale to better show the correlation at other sites.

Figure 5: Put the JPL r2 values on each graph.

Figure 5c: Ethene vs ethane seems to have a natural gas wing. Same with the light green data (New Mexico). Just interesting.

Figure 8 needs stronger fonts.

---

## Author Comment (AC1) · 29 Sep 2017

Review of the paper "Measurements of atmospheric ethene by solar absorption FTIR spectrometry" and author responses (in blue)

The authors thank the reviewer for their most careful and thorough review. We appreciate the considerable time that this must have taken.

Anonymous Referee #1
This paper describes the retrieval of atmospheric ethene amounts from 30 years of solar remote sensing measurements from the ground and stratospheric balloon. Ethene absorbs only weakly in the spectra, and the analysis is very careful, comprehensive and reliable. The measurements are described in detail, and compared to a range of other published remote sesnig and in situ measurements. The conclusions focus on the observed decrease in C2H4 amounts over the period, but the discussion of reasons for this decline is somewhat disjointed and anecdotal. A restructure of the discussion and conclusions sections would improve the paper and do justice to the high quality measurements. The paper is suited to ACP readers and I recommend publication after minor revisions listed below.
Thank you.

L 11 ethane should read ethene
Fixed

L 11 1990s not 1990's
Fixed

L 17-18. Suggest rephrasing without "etc." for example "Atmospheric ethene is formed primarily by incomplete combustion from sources such as biomass burning, power plants and combustion engines."
Done.

L 17-25 Ambiguity: Sawada and Totsuka are referenced 3 times, redundantly, together with Goldstein, but it is not clear if the referenced fluxes are all fluxes or just biogenic. Please rephrase.
Removed one of the Sawada and Totsuka references. Hard to remove another because the first use is for their ethene emissions, whereas the second use is for their ecosystem areas. The second reviewer wanted more serious revisions to the Introduction, which have somewhat over-run these changes.

L 31 Not all readers will be familiar with "MkIV" - briefly describe "MkIV" at first use, for example ". . . with the remote sensing results described here from the MkIV solar infrared interferometer.
I have added "an infrared Fourier transform spectrometer that uses the sun as a source."

L44 The website reference may change in the future. Could a snapshot or table of the webpage be added as an appendix or supplementary info to the paper?
Good point. The other reviewer also complained about this. I have added a new Table in Supplementary Information.

L 55/Fig 1 The red C2H4 and H2O spectra are almost impossible to distinguish, please choose a colour that makes C2H4 stand out. It is in fact clearer in Figure 2, but the reader may not realise this.

I have remade figs 1 & 2 will more color separation between the C2H4 (red) and H2O (orange). But in doing so, the CO2 is now more yellow, which is a hard-to-see color. The fundamental problem is that in a figure containing 10 colors (11 if you include black), it is difficult to make them all clearly distinct from each other.

L107 This text is exactly repeated in the Figure 2 caption. It could be left out of one or the other.
Done. Omitted from caption.

L125 The solid lines in Figure 3 show. . . (not shows. . .)
Done.

L 148 The names and locations of the 12 sites should be provided here or in a table. Many (most) readers will not be familiar with the MkIV sites.
Added a reference here to the new table in Supplemental Information.

L177 "Where" or "While" rather than "Whereas"?
Changed sentence to: "In the 1990's $C_2H_4$ often topped $15x10^{15}$ molec.cm$^{-2}$, but since 2010 a column exceeding $7.5x10^{15}$ has only been observed once."

L 184 et seq. This statement is out of place here – it is more in the nature of discussion than presentation of results, and represents the authors' opinion without real quantitative backup. This sentence and other similar points in the results section may be better collected in the discussion section.
Moved it to the Discussion.

L198 have => having
Agreed.

Figure 6 The colours are different from those in Fig 1 & 2, can they be made consistent (with C2H4 easy to distinguish from other gases).
Good idea.

L 296 spell out AOD and MIR
Done.

L311 Figure 2 of ???? et al.
Strange.  Says Herbin in the MS Word document, but is blank in PDF. Font problem?

L 319 Is it possible to ask Herbin or co-author rather than speculate about the log retrieval?
I asked Chris Boone of the ACE science team and a co-author on the Herbin paper. Chris confirmed my suspicion that a log(vmr) retrieval was performed. So I have therefore changed the text from "speculate" to "believe".

L321 retrieval mis-spelt
Fixed.

L335 Ethene measurements, not ethane
Fixed.

L337 lower case P in Profiles Discussion
Fixed.

L404-417. This paragraph is a little frustrating – do the authors believe that the urban areas are NOT responsible for the high CO and C2H4, or that the trajectories are inaccurate ("wrong" is a strong word). Having mentioned both possibilities, the discussion is left hanging without any conclusion.

I completely agree. And I was hoping that I would discover some kind of an error in my use of the Hysplit tool by now.  But this hasn't happened.  So what to do?  Omit the paragraph and pretend that I never did a Hysplit analysis, or own up to a negative result?  This paragraph has been moved into Methods at the suggestion of the other reviewer.

L418-424 There a few points scattered through the results pointing towards the secular decrease and its causes that could be pulled together here into the discussion, which is currently rather light. Some of this discussion currently resides in the summary and conclusions – I would recommend expanding the discussion and reducing the Conclusions to the main points of the study

Okay. But the other reviewer has requested moving some material out of Discussion and into Methods.

---

## Author Comment (AC2) · 29 Sep 2017

Reviews of the paper "Measurements of atmospheric ethene by solar absorption FTIR spectrometry" and author responses (in blue)

The authors thank the reviewers for their most careful and thorough reviews. We appreciate the considerable time that this must have taken and can say without hesitation that this is the most thorough review that we have encountered in 30+ years.

Reviewer's comments in black. My responses in blue.

Anonymous Referee #2
This paper reports measurements of atmospheric ethene columns at a dozen sites. Downward trends were observed in Pasadena, likely related to regulations on vehicle emissions. This paper presents interesting information and should be published, but much of the discussion is too speculative and needs to be strengthened. Basic information about the sampling sites/times and instrument performance is needed.
A new Supplemental Information appendix adds site information. Basically, it is a screenshot of the tables on the webpage: http://mark4sun.jpl.nasa.gov/ground.html, which have been recently updated to include information about the season and the time of day of the measurements, plus the type of terrain that surrounds each site.

The impact of an apparently high detection limit on the results needs to be clarified,
A new Supplemental Information appendix adds information about the uncertainties associated with the MkIV C2H4 measurements. And a new paragraph discusses this appendix.

for example is lack of seasonality or "essentially no" C2H4 more related to the instrument rather than actual lack of seasonality?
At JPL, where C2H4 is usually above the detection limit, the lack of seasonality apparent in figure 4b is a real feature. At the clean sites these data can't say anything about the seasonality.

The presentation also needs some clarification and restructuring to make it easier for the reader to follow.
Okay.

General comments: Abstract: The abstract could be filled out by stating roughly where the dozen sites are located,
I've added "mostly between 31N and 68N"

how many sites were clean-air vs urban, how many sites fell below the detection limit, and how many sites showed measurable trends. Currently only 4 clean air sites and one urban site are discussed.
I think that this is too much detail for the abstract. But I have added a new table in Supplemental Information that addresses many of these issues.

General comment: There is a lot of colloquial language in the paper (L56, L65, L90, L335, L393, etc.). Use specific scientific language.
Fixed.

Introduction: The first sentence seems strange because it misses the biogenic source.

Agreed. I have now remedied this by changing the first sentences to "Atmospheric ethene is formed by microbial activity in soil and water, biological formation in plants, and by incomplete combustion from sources such as biomass burning, power plants, and combustion engines."

What is the partitioning of combustive vs biogenic ethene in its global budget? If the biogenic source is 21 Tg/yr, how large is the combustive source?
Have added a sentence referencing Abeles et al. (1992), who assert a Terrestrial biogenic source of 16.6 MT/y and an Anthropogenic source of 9.2 MT/y.

Were Poisson et al. measuring biogenic ethene? It's not in the references.
As I understand it, this is a 3D model prediction -- so they weren't measuring anything.  Have added Poisson 2000 to references.

Overall the first paragraph needs tightening up.
Agreed.

L65: "This is a "greatest hits" compilation . . . not always the latest version for every band of every gas". This is too vague . . . be more specific in describing the limitations
There are already three sentences that follow the quoted sentence that provide additional specificity. Without knowing what additional information you would like, I don't know how to proceed.  I could write a dozen pages on the linelist and still might not address the specificities that you desire.

 give some sense of your overall precision and how this varied over the years or at different sites.
I have added the following paragraph discussing the C2H4 measurement uncertainty.
"The main limitation to the precision of $C_2H_4$ measurements by the solar absorption technique is the ability to accurately account for the absorption from $CO_2$, $H_2O$, and $SF_6$, which overlap the $C_2H_4$ Q-branch.  The first two gases, in particular, being much stronger absorbers than the $C_2H_4$, have the potential to drastically perturb the $C_2H_4$ retrieval.  For example, an error in the assumed $H_2O$ vmr vertical profile, and hence the shape of the $H_2O$ absorption line, will have a large affect on retrieved $C_2H_4$.  And since the overlapping $CO_2$ lines are so T-sensitive, a small error in the assumed tropospheric temperature will also greatly influence the $C_2H_4$ retrieval.  Errors in the spectroscopy of $H_2O$ and $CO_2$ will also affect $C_2H_4$ retrievals.  Figure S.2 shows the $C_2H_4$ retrieval uncertainties, estimated from the spectral residuals, together with jacobians of the various retrieved quantities.  The same data are plotted versus year, solar zenith angle and site altitude.  From JPL the measurement uncertainty is about $0.5E+15$ molec.$cm^{-2}$.  At higher solar zenith angle the uncertainty decreases as the $C_2H_4$ absorption deepens.  At higher altitudes the uncertainty decreases as the interfering absorption decreases faster than that of $C_2H_4$.  There has been no significant change in the $C_2H_4$ retrieval uncertainty over time.  We note that the measurements made from McMurdo Antarctica in Sep/Oct 1986 have very small uncertainties, due to their high airmass and the extremely small $H_2O$ absorption.  The plotted uncertainties represent a single observation representing a 10-15 minute integration period."

L95: Before getting into specific Pasadena results we need a Table stating the 12 sites, their latitude, longitude and altitudes, and the dates over which measurements were taken, what time of day and for how long, whether various sites were considered clean or polluted, etc. For example the "Ground-based Sites and Observations (1985- 2015)" table from http://mark4sun.jpl.nasa.gov/ground.html (L44) could be included, with sample number and dates added in (observation days are only listed until 2004 on the website).
I have added Supplemental Information containing updated versions of the tables on the website, in case at some point in the future the link disappears.  It contains most of the stuff that you asked

for. I have added a one-word description of the terrain surrounding each site and the season and time of day of observations.

Regarding whether the site is clean or polluted, this varies on a day-to day-basis. While the urban sites are generally polluted, this is not always the case. For example, when the winds are from the Northern quadrant, which happens frequently in the winter, JPL can be a clean site. And in all of the non-urban sites, I have seen pollution events. So there isn't a clean division between polluted and clean sites.

L152: We still don't know basics such as the precision of the instrument (I'm assuming the detection limit is 10^15 molec.cm2 based on the abstract, but it's not clearly stated).
I have added a paragraph (above) discussing the C2H4 measurement uncertainty, and a figure in Supplemental Information.

What is meant by "uncertainties" here, and what is the basis for choosing
As stated in the new paragraph, " retrieval uncertainties are estimated from the spectral residuals, together with jacobians of the various retrieved quantities. The uncertainties are then the square root of the diagonal elements of the resulting covariance matrix. This is the standard Optimal Estimation procedure.

L162: "but never come close to zero". Why is this stated?
To contrast the CO with the other gases in fig. 4, which have many measurements near zero.

CO has a non-zero background concentration, so it wouldn't be expected to approach zero.
Agreed. And it doesn't.

On L181 the statement that the CO decline is not as dramatic as ethene because it doesn't fall as low doesn't make sense for the same reason. By what factor did CO fall relative to its background, and how does that compare to ethene?
It is evident from figure 5a that there is a linear relationship between CO and C2H4, but it does not go through the origin. If I were to *assume* that the background CO is 1.5E+18, and subtract this from the CO columns, then the fractional decrease in CO would of course then match that of C2H4. But how do I prove that the CO background is truly 1.5E+18? What if there are no clean days at JPL?

What is the clean background value for CO at Pasadena latitudes in winter and summer?
Don't know. Pasadena is never completely clean in the summer.

L169: Where is the TMF site relative to LA (what direction)?
TMF is almost due East from JPL, but is 2 km higher in altitude

Were winds from a clean or polluted direction?
There are 45 days of observations from TMF, so it is hard to generalize. In any case, the wind direction isn't a good indictor of pollution over TMF. Even if the winds are blowing directly from the SW, the direction of Los Angeles, if the PBL thickness is < 2 km, then the Los Angles pollution will be trapped by the mountains and inversion. In such situations TMF will receive clean air from the Pacific. In general, in the mornings TMF is a clean site. In summer afternoons TMF can become polluted as the PBL thickens over Los Angeles, allowing the pollution to reach the 2.3 km altitude of TMF. But our measurements, being in the autumn, have never seen this.

On L168 this is the first time we're told what season the measurements were made.
The new Supplemental Material contains information about the seasons and time of day of observations.  But these are general statements.

Figure 4 needs a legend to match color with site . . . which site goes with which altitude?
Panel 4a shows the relationship between color and altitude, which I thought would eliminate the need for a legend.

Same with Figure 5. Though altitude may not be the best way to color-code . . . places as different as McMurdo and Texas come out as the same color and the 12 sites can't be distinguished.
I agree that altitude is not a perfect way to color-code the figure. I have tried latitude, longitude, but always get sites that cannot be distinguished.  Color-coding by altitude is the least bad method that I have discovered so far.

L170: "The only sites where MkIV has ever detected C2H4. . ." What color is Sweden and what season was it measured (winter)?
Esrange, Sweden, is at 0.27 km altitude and therefore appears as light blue.   But measurements from Sweden were made only in 1999/2000, 2002/3, and 2006/7.

There seems to be some light blue around 1989 that is measurable?
I don't see any light blue in 1989.  As shown in the updated table S.2, measurements in 1989 were from Ames Research Center (ARC; dark blue), JPL (green), and FTS (lime-green).

In each panel of Figure 4, a line should be drawn indicating the detection limit.
The detection limit is site-dependent and zenith angle dependent, as illustrated in the new Fig S.3. This maes it difficult to draw a single line to represent all measurements.

Line 175: The evidence for this is a little weak . . . a wind rose plot would clearly show how the ethene levels vary with wind direction and how much of the large range is related to wind direction versus other things like time of day or seasonality.
A wind rose is a 2-D construct.  In mountainous terrain the third dimension becomes important. As I said earlier, a wind direction from Los Angeles will not bring pollution to TMF if there is a strong inversion over the LA basin.  It will bring clean Pacific air that crossed the LA basin at 2.3 km altitude.

L183: The arguments in this paragraph, while most likely correct, are too speculative. Apart from the Clean Air Act of 1990, when did CARB policies and stronger enforcement of smog checks occur?
I have added references to Bishop and Stedman (2008).
Also, the website of the California Bureau of Automotive repair shows a 24-step chronology of the California Smog Check, from 1972 to 2014:
www.bar.ca.gov/FormsPubs/Fact_Sheets_and_Brochures/Smog_Check_Program_History.html
And Wikipedia also has an informative article:
https://en.wikipedia.org/wiki/California_Smog_Check_Program
But I see no reason to regurgitate this information into the paper.

Is the ratio of C2H4/CO consistent with traffic? If you believe it's traffic, why is biomass burning raised as a possibility on L422-423?
I have removed the speculation about biomass burning

It may help to discuss the Sather and Cavender and Washenfelder et al. papers here rather than below.
The other reviewer wanted the sentence residing line 184-189 moved to the Discussion, which I have done. This being the case, there is no longer any point in discussing the Sather and Cavender and Washenfelder et al. papers here.

L196: What do you mean by "not as tight"? What is the r2?
Changed "tight" to "compact". I don't think that r2 is a useful statistic in a plot showing data from multiple sites, each of which has a different gradient.

What other sources do you expect ethyne to have in an urban center?
I don't know. Perhaps oxy-acetylene welding (fugitive emissions)?

Unlike ethane it's also a combustion tracer like ethene, so it should behave like CO and ethene.
And it does.

What does the correlation between CO and ethyne look like?
It looks very good. This was plotted in the Supplementary Information. Table 1 showed that the Pearson correlation coefficient was 0.91.

L205: Why "seems"?
Deleted.

What does statistical analysis show?
That's what Table 1 is all about: the gradients, their uncertainties, and correlation coefficients of the gas relationships. Are you asking me to summarize the results of Table 1?

Do not use wording like "large values of the red points in the third row". Use scientific descriptions.
Now states "$C_2H_6$ has not decreased significantly as is evident from the third row of Fig. S.1, which shows that the 2015 column abundances (red) span similar values to those measured in 1990 (blue)."

L230: Since the ratio appears to be changing over time, what is you ratio for 1999-2005, the same time-frame as Baker et al.? That would be better than comparing 1985-2016 to 1999-2005.
Good point. Have modified this paragraph to: " The overall gradient of the $C_2H_4$/CO relationship using all JPL data is 3.7±0.4 ppt/ppb, as in Table 1, but the post-2010 data have a gradient of only 2.7±0.4 ppt/ppb. Baker et al. [2008] measured $C_2H_4$/CO emission ratios of 5.7 ppt/ppb in Los Angeles from whole air canister samples acquired between 1999 and 2005, which is close to their average of all US cities, 4.1 ppt/ppb. Over this same time period the MkIV JPL data reports 4.0±0.7 ppt/ppb, the larger uncertainty reflecting the relatively few observations from JPL over this period. Warneke et al. [2007] report a $C_2H_4$/CO emissions ratio of 4.9 ppt/ppb in Los Angeles in 2002, measured by aircraft canister samples."

L334: Suggest condensing this section. It's sort of a laundry list of other projects without much synthesis.
I agree: it's a laundry list. I will try to condense and add a little synthesis.

On L338 if SE Asian measurements are not relevant to this study, why are they presented (e.g., Blake et al., 2003; Figure 8)?

The text on L338 states "**mainland** SE Asia".  I don't want to compare super-polluted measurements from China.  But aircraft measurements over the Pacific will be much less affected by local sources and so provide something to compare with MkIV balloon measurements.

Probably a Table would be a better way to intercompare results and show the different years, seasons and locations of each campaign.
Are suggesting replacing figure 8 with a table?  I prefer to keep the figure and add the requested info into the text, which is what I have done.

L381: In comparing the mid-Pacific to the USA, you need to state what year and season the different missions flew and what impact this might have had (the figure is comparing winter/spring flights from 1994-1999 with HIPPO over a decade later). What phase of HIPPO is plotted?
I'm plotting HIPPO data from a file named "HIPPO_discrete_continuous_merge_20121129.tbl". The data over the mid-pacific at 30-40N were acquired in Jan & Dec 2009, Apr 2010, and Jul 2011.  The data over the US were acquired in Jan & Dec 2009, and Jul 2011. This is now sated in the paper.

Are the mid-Pacific and USA HIPPO data from the same season?
Have added the sentence " The cyan points were measured mid-Pacific in Jan and Dec 2009, Apr 2010, and Jun/Jul 2011.  The green points were measured over the Central/Western USA in Jan and Dec 2009, and Jun/Jul 2011"

Is L381-382 referring only to the HIPPO data (which is blue and green in the figure; no red points)?
Yes.  I stated the colors incorrectly. I have now added the sentence ".  Profiles from the PEM-West B and TRACE-C aircraft campaigns are also plotted in red and blue."

On L384 what evidence was there for upward transport of Asian pollution to high altitude? Is this referring to HIPPO data?
The evidence is in the shape of the HIPPO profiles. The mid-Pacific profiles decrease more rapidly with altitude than those over the US.  Upward transport would act to reduce the altitude gradients.

this paragraph needs tightening and better links to the rest of the paper.
Agreed.

L409: Use less speculative arguments . . . "tends to discount the possibility that the C2H4 measurements are wrong" is not convincing. Suggesting that the trajectories are wrong or that urban pollution isn't a major source of ethene or CO probably isn't the direction you want to go.
Deleted " or that the urban pollution is not a major source of the $C_2H_4$ or CO observed by MkIV"

Do you get better correlations when you remove data originating from the San Gabriel direction?
This is sort-of what I was trying to do with the Hysplit analysis.  But it didn't pan out the way I had hoped.

L418-423 is too speculative. No evidence was provided for decreasing emissions from biomass burning in the Pasadena area . . . if you believe this is the case your argument needs to be much more substantial.
Agreed. Have removed the sentence " One possibility is decreasing pollution from biomass burning, which causes higher $C_2H_4$/CO emission ratios [Lewis et al., 2013]."

Minor corrections/clarifications:

General: I suggest numbering your sections so they fall more clearly into Methods, Results etc.
Agreed and done.

L7: All acronyms need to be defined, even if they seem obvious. Define JPL.
Defined JPL (twice), FTS.

On L22 define PAR, and so forth (L98, L120, L220, L296).
Defined PAR on line 22.

L11: Ethane should be ethene. Same on L335 and L346.
All fixed

L50: I don't see a black line in the lower panel.
The black line is present in fig.1. But the fit is so good that it is buried beneath the black points.
The black line is visible in figure 2 thanks to the zooming.

L52: Many of these terms (continuum level, etc.) might not mean much to the average reader.
Changed to " *Also fitted are the 0% and 100% signal levels, separate telluric and solar frequency shifts, together with 5 more weakly absorbing gases ($NH_3$, $SF_6$, $COF_2$, $O_3$ and $N_2O$)."*

L56: "is less than 1% deep" . . . use more specific wording or define deep.
Changed to ". *The $C_2H_4$ absorption contribution (red) peaks at 949.35 $cm^{-1}$ with an amplitude of less than 1% and therefore difficult to discern on this plot"*

L57: 0.235% is a very precise number . . . is this the level of significance you intend?
Yes, because I subsequently compare this with 0.251%, which is the RMS when C2H4 is excluded from the fit. The difference between 0.235 and 0.251 is significant, even though the absolute value of each number isn't.

L58: Use spaces to indicate minus rather than a hyphen in "measured-calculated".
Not sure what you mean. I have inserted spaces before and after the minus.

L77: Is this what you used? Just this sentence is out of place without some link to your study.
Changed to "We analyzed the strongest infrared absorption feature of ethene: the Q-branch of the $v_7$ band ($CH_2$ wag) at 949 $cm^{-1}$."

L95: Give an exact lat/long and describe the site. In a field? Near a road? Wind direction?
Added the sentence "For data acquisition from JPL, the MKIV instrument was indoors with a coelostat mounted to the south wall of the building feeding direct sunlight into the room." The exact lat/longitude is now in Supplementary Information Table S.2

L102: State which day. The caption just says March 2014.
Added March 17, 2014.

L107-110: This is almost the same wording as the caption.
The fig.2 caption has already been shortened at the request of Reviewer 1

L125-126: Similar wording as the caption.
Shortened sentence to " Figure 3 show the averaging kernel and a priori profile pertaining to the $C_2H_4$ retrieval illustrated in Figures 1 and 2", which is less similar to caption.

L131: Text uses ppt. Is the x-axis of the lower panels also ppt? Units aren't given and the text is very difficult to read because of scientific notation. Just use ppt and a scale from 0 to 500.
Good idea. Figure has been remade.

L136-148: This is more Methods than Results.
Agreed. And the previous paragraph too. So I moved the "Results: Ground-based MkIV Retrievals" section header to later in the paper.

L144: "smallness" is a strange word. How about just "This small C2H4 column perturbation. . ."
Agreed and fixed.

L152: Units are needed for 1x10^15.
Done.

L164: Define TMF. This site designation doesn't mean anything to the reader.
TMF is now defined at first use.

L171: What was the season, time of day and wind direction for the Mountain View measurements?
July 87, Oct 91 and Dec 91. All around noon. Don't know the wind direction.

L173: No capital for Northern.
Fixed.

L181: Change "The CO" to "CO" or "The CO column". Same on L203.
Done.

L183: State the r2 value after "tight correlation".
I've stated the Pearson correlation coefficient (PCC). I believe that the r2 is the square of the PCC. I prefer the PCC because it spans -1 to +1, whereas the r2 spans 0-1.

L193: "poorer" . . . how poor compared to 0.93?
Added "*as low as 0.30 for $H_2CO$*"

How about Mountain View, the other urban site?
There are only 7 observations on 4 different days from Mountain View.

L198: Should be having not have.
Fixed.

L204: Awkward wording.
Changed sentence to "In the 1990's $C_2H_4$ often topped $16x10^{15}$ molec.cm$^{-2}$, but since 2010 a column exceeding $8x10^{15}$ has only been observed once."

L208: 2016 not 2916.
Fixed.

L208: Typo . . . "a 2.5% enhancements". Similar issue on L243.
Fixed.

L224: An emission ratio subtracts off the background. Was that done here for CO?
Yes. This happens implicitly when you fit a straight line (gradient & intercept) to a plot of gas column vs CO column.

L229: What is the distinction between JPL ground-based data only (Table 1) and "all JPL data"?
No real distinction, just different ways of saying the same thing. The "all JPL data" refers to the entire 1985 - 2016 time period, rather than the post-2010 subset mentioned in the same sentence.

L231: Did Baker et al. and Warneke et al. report uncertainties?
Warneke (2012) show a figure with an error bar that I estimate to be about +/- 10% for the Delta_C2H4 / Delta_CO emission ratio.

L263: Why 949.4 here but 949.35 on L246?
Changed to 949.35

L272: Use consistent units. The text cites 65 ppt but Figure 7's x-axis uses 6x10-11. Just use ppt.
Good idea. Re-made figure in ppt

L297-298: Not sure of the point of this statement relative to your paper.
Deleted.

L305: Not necessary to include "their Table 2" and "Their Fig. 2".
Removed.

L311: Correct "of et al." The paragraph needs some re-writing . . . it's too casual.
Strange. Says "of Herbin et al." in the MS Word document, but "Herbin" is missing in PDF.

L314: Why "presumably"? Is it not clear from the paper? Same on L320 and L342.
No, it is not clear from the Herbin paper. I had to ask a co-author (Chris Boone) to get the full story.
Removed the first "presumably". Changed the "speculate" to "believe". Left the second "presumably".

L321: Typo, retrieval.
Fixed.

L335: Too colloquial. This paragraph needs proofing.
Proofed and modified.

L347: 1848 is too precise for an average; add an error bar if they had one
I agree, but that's the value they report and I don't feel at liberty to round it.

Same for on L348 for the ER of 6.97.
Again I agree with you, but don't want to modify their published values

L351: Typo, "fig.S1".
Not a typo. This is how I reference the figure in Supplemental Information

L368: No comma after measurements.
Fixed.

L377: HIAPER not HAIPER.
Fixed.

L381: I think you mean "green squares" rather than "red points". Same on L382.
Yes, I got the colors wrong. Fixed.

L386, L394: No comma after al.
Fixed.

L387: No hyphen for precursor.
Fixed.

L399: When were the measurements in Mexico City? Are they ground-level?
1999, 2002, 2003.  Yes, ground level.

L404: This is methods/results more than discussion.
Agreed. It has been moved. Which makes the Discussion section even thinner.

The conclusions read more like a summary.
The section is titled "Summary and Conclusions"

References need consistent formatting.
Yes.

Table 1: The significant figures in the gradient and error need to match: 1.3 +/- 0.1 but not 1.28 +/- 0.1 and so forth.
Done.

Figure 1: Tidy up the graph for publication (stronger font, less writing on top – or if you include it define all the symbols).
I don't know how to make the font any stronger.  I have used the same font for 20+ years without complaint.  If ACP wants me to make the font darker I will do it, but this will take time.
I have expanded the figure caption to define the numbers at the top.

In Figures 2 and 6 the writing on top is cut off.
Removed in fig 2.  Fixed in Fig.6.

Figures 1 and 2: Is the top panel in Figure 1 the same as the middle panel in Figure 2? If so delete the top panel in Figure 1. The two figures could probably be merged.
Figure 2 is a zoom into fig.1, to allow the $C_2H_4$ absorption can be seen.  The top panel of fig 1 cannot be deleted because it contains information outside the range covered by the middle panel of fig.2

Figure 3: Axis labels are not clear (are overlaid). Need a larger font.
Changing to ppt removed the overlap. Increased font slightly.

Figure 5 could be on a log scale to better show the correlation at other sites.

The negative column amounts would then all be lost. And the very small column values would also be lost because these would have large negative values of log(column) and would have to be clipped. I like the fact that the linear scale shows that the column amounts at the clean sites are equally positive and negative.

Figure 5: Put the JPL r2 values on each graph.
This would be misleading because figure 5 shows ground-based column results from all sites, not just JPL. Figure S.1 shows the JPL-only column results. [Sorry for the confusion. Fig S.1 used to be in the main paper, it was moved to Supplementary Information. Table 1 relates to fig. S.1, not fig.5].

Figure 5c: Ethene vs ethane seems to have a natural gas wing. Same with the light green data (New Mexico). Just interesting.
Agreed. There are many natural gas leaks in the LA basin, the Aliso Canyon leak in particular. This natural gas wing is more obvious in the fig. S.1, which shows the JPL-only results.

Figure 8 needs stronger fonts.
I don't immediately know how to do this.

---

## Author Response (AR2)

**Author Response to Editor's Minor Comment (Dec 5, 2017)**

There is one issue raised by Reviewer 2 that I think the authors need to clarify and possibly reconsider. This is the discussion about changes in the ratio of C2H4 and CO at JPL over time which is in lines 212-216 and the interpretation of Fig 5a and part of Table 1.

Okay.

The main point is that it is not clear exactly what the authors are calculating: (a) is the correlation coefficient calculated on the raw measurements or after the sample means are taken out?;

The latter. The manuscript clearly states that we used the Pearson Correlation Coefficient (PCC). This is defined by the equation

$$r = \frac{\sum_{i=1}^{n}(x_i - \bar{x})(y_i - \bar{y})}{\sqrt{\sum_{i=1}^{n}(x_i - \bar{x})^2}\sqrt{\sum_{i=1}^{n}(y_i - \bar{y})^2}}$$

which subtracts the means from the x- and y-values.

(b) is the gradient calculated on a line forced through zero or is an non-zero intercept allowed?

The latter. I have modified the text to make this clearer.

If a non-zero intercept is allowed, then the discussion about the background C2H4 is moot, as an effective background of about 1.5E18 for CO is found from the data.

Agreed. My response to Reviewer 2's comment that: "L224: An emission ratio subtracts off the background. Was that done here for CO?" was: "Yes. This happens implicitly when you fit a straight line (gradient & intercept) to a plot of gas column vs CO column."

What is then interesting is if the slope of lines - or the intercepts - calculated for different time periods changing as that implies a change in the emission ratio.

Agreed. This is why I devote two paragraph tos discussing this.

So my recommendation is that the authors (a) clarify exactly what they have calculated in the text and in Table 1, e.g. by drawing a fitted line in Fig5a;

Oh dear, there seems to be a misunderstanding.  The gradients and correlation coefficients in Table 1 refer to Figure SI.1, not Figure 5.  Reviewer 2 was also confused about this.  I am not adding straight lines, derived from JPL-only data in figure SI.1, to figure 5, which includes data from all sites.

I think that the underlying problem here is that the most-discussed figure in the whole paper is currently languishing in Supplementary Information, after being relegated from the main paper in June. But the discussion of it remains in the main paper.

So I have merged former figure SI.1 with Table 1.  This was achieved by writing the gradients and correlation coefficients into each panel of Fig. SI.1.  I have also added the fitted straight lines.  The resulting Figure 6 is in the main paper, where Table 1 used to be.  Figures 7 & 8 have been re-named to Figures 8 & 9.

Also, the old SI.2, which contained two tables (SI.2a and SI.2b), has been split.  SI.1 now contains the list of sites and their properties, while SI.2 contains the number of observation days at each site in each year.  SI.3 is unchanged.

Minor:
134: Figure 3 shows....

Fixed.